# A global mean sea-surface temperature dataset for the Last Interglacial (129-116 kyr) and contribution of thermal expansion to sea-level change

Chris S.M. Turney[1,2]*, Richard T. Jones[3]†, Nicholas P. McKay[4], Erik van Sebille[5,6], Zoë A. Thomas[1,2], Claus-Dieter Hillenbrand[7], Christopher J. Fogwill[1,8]

[1]Palaeontology, Geobiology and Earth Archives Research Centre, School of Biological, Earth and Environmental Sciences, University of New South Wales, Australia
[2]ARC Centre of Excellence in Australian Biodiversity and Heritage, School of Biological, Earth and Environmental Sciences, University of New South Wales, Australia
[3]Department of Geography, Exeter University, Devon, EX4 4RJ, UK
[4]School of Earth and Sustainability, Northern Arizona University, Flagstaff, Arizona 86011, USA
[5]Grantham Institute & Department of Physics, Imperial College London, London, UK
[6]Institute for Marine and Atmospheric Research Utrecht, Utrecht University, Utrecht, Netherlands
[7]British Antarctic Survey, High Cross, Madingley Road, Cambridge CB3 0ET, UK
[8]School of Geography, Geology and the Environment, Keele University, ST5 5BG, UK
†Deceased.

*Correspondence to*: Chris Turney (c.turney@unsw.edu.au)

**Abstract.** A valuable analogue for assessing Earth's sensitivity to warming is the Last Interglacial (LIG; 129-116 kyr), when global temperatures (0 to +2°C) and mean sea level (+6 to 11 m) were higher than today. The direct contribution of warmer conditions to global sea level (thermosteric) are uncertain. We report here a global network of LIG sea surface temperatures (SST) obtained from various published temperature proxies (e.g. faunal/floral assemblages, Mg/Ca ratios of calcareous plankton, alkenone $U^{K'}_{37}$). We summarise the current limitations of SST reconstructions for the LIG and the spatial temperature features of a naturally warmer world. Because of local $\delta^{18}O$ seawater changes, uncertainty in the age models of marine cores, and differences in sampling resolution and/or sedimentation rates, the reconstructions are restricted to mean conditions. To avoid bias towards individual LIG SSTs based on only a single (and potentially erroneous) measurement or a single interpolated data point, here we report average values across the entire LIG. Each site reconstruction is given as an anomaly relative to 1981-2010, corrected for ocean drift and where available, seasonal estimates provided (189 annual, 99 December-February, and 92 June-August records). To investigate the sensitivity of the reconstruction to high temperatures, we also report maximum values during the first five millennia of the LIG (129-124 kyr). We find mean global annual SST anomalies of $0.2 \pm 0.1$°C averaged across the LIG and an early maximum peak of $0.9 \pm 0.1$°C respectively. The global dataset provides a remarkably coherent pattern of higher SST increases at polar latitudes than in the tropics (demonstrating the polar amplification of surface temperatures during the LIG), with comparable estimates between different proxies. Polewards of 45° latitude, we observe annual SST anomalies averaged across the full LIG of $>0.8 \pm 0.3$°C in both hemispheres with an early maximum peak of $>2.1 \pm 0.3$°C. Using the reconstructed SSTs suggests a mean LIG global thermosteric sea level rise of $0.08 \pm 0.1$ m and a peak contribution of $0.39 \pm 0.1$ m respectively (assuming warming penetrated to 2000 m depth). The data provide an important natural baseline for a warmer world, constraining the contributions of Greenland and Antarctic ice sheets to global sea level during a geographically widespread expression of high sea level, and can be used to test the next inter-comparison of models for projecting future climate change. The dataset described in this paper, including summary temperature and thermosteric sea-level reconstructions, are available at https://doi.pangaea.de/10.1594/PANGAEA.904381 (Turney et al., 2019).

## 1 Introduction

The timing and impacts of past, and future, abrupt and extreme climate change remains highly uncertain. A key challenge is that historical records of change are too short (since CE 1850) and their amplitude too small relative to projections for the next century (IPCC, 2013;PAGES2k Consortium et al., 2017), raising concerns over our ability to successfully plan for future change. While a wealth of geological, chemical, and biological records (often referred to as 'natural archives' or 'palaeo') indicate that large-scale and often multi-millennial duration shifts in the Earth system took place in the past (Thomas, 2016;Steffen et al., 2018;Lenton et al., 2008;Thomas et al., 2020), there are limited global datasets of such events. A comprehensive database of environmental

conditions during periods of warmer-than-present-day is essential for constraining uncertainties surrounding projected future change, including sea level rise, extreme weather events and the climate-carbon cycle. In this regard, the Last Interglacial (LIG), an interval spanning approximately 129,000 to 116,000 years ago, is of great value (Dutton et al., 2015). Described as a 'super-interglacial' (Turney and Jones, 2010;Overpeck et al., 2005), the LIG was one of the warmest periods of the last 800 kyr, experiencing relatively higher polar temperatures compared to the global mean ('polar amplification') (Past Interglacials Working Group of PAGES, 2016;Hoffman et al., 2017;Turney and Jones, 2010;Capron et al., 2017), with the most geographically widespread expression of high global mean sea level in the recent geological record (GMSL, +6.6 to +11.4 m) (Dutton et al., 2015;Grant et al., 2014;Kopp et al., 2009;Rohling et al., 2017), abrupt shifts in regional hydroclimate (Wang et al., 2008;Thomas et al., 2015), and elevated atmospheric $CO_2$ concentrations (relative to the pre-industrial period) of ~290 ppm (Köhler et al., 2017;Schneider et al., 2013;Barnola et al., 1987;Petit et al., 1999), suggesting non-linear responses in the Earth system to forcing (Steffen et al., 2018;Thomas, 2016;Dakos et al., 2008;Thomas et al., 2020). Importantly, there remain considerable debate over the contribution of sources to the highstand in global sea level (Dutton et al., 2015;Rohling et al., 2019). Previous work has suggested ocean thermal expansion contributed some 0.4 m (McKay et al., 2011), while Greenland Ice Sheet melt is estimated at some 2 m (NEEM Community Members, 2013) and melting mountain glaciers ~0.6 m (Dutton et al., 2015), implying Antarctic mass loss >3.6 m (Fogwill et al., 2014;Turney et al., 2020;DeConto and Pollard, 2016;Dutton et al., 2015;Rohling et al., 2019). Constraining the different contributions to GMSL during the LIG requires a comprehensive ocean temperature database to precisely quantify the role of ocean thermal expansion, compare to climate model-generated temperature estimates, and use these temperature estimates to drive ice sheet models (Fogwill et al., 2014;Mercer, 1978;DeConto and Pollard, 2016;Sutter et al., 2016;Hoffman et al., 2017;Clark et al., 2020).

Quantified temperature reconstruction data for the LIG are often drawn from disparate publications and repositories (usually reported alongside other Late Pleistocene data). To obtain reliable temperature reconstructions, it has until recently proved necessary to determine a global estimate of the magnitude of warming using only a selected number of "high-quality" records; the resulting temperature reconstructions of LIG temperatures ranged from 0.1 to >2°C warmer than present (CLIMAP, 1984;White, 1993;Hansen, 2005;Rohling et al., 2008;Turney and Jones, 2010). With the ever-increasing number of quantified temperature reconstructions of the LIG reported in individual publications, it is crucial that these datasets are brought together to derive a comprehensive reconstruction of global change during the LIG. A further consideration is that in contrast to terrestrial sequences, marine records typically provide a continuous record of LIG conditions (Turney and Jones, 2010;Turney and Jones, 2011), providing an opportunity to determine the sensitivity of GMSL to sea-surface temperature (SST) conditions during the interglacial (including early maximum temperatures). Given a possible warming of 2°C (Turney and Jones, 2010;Fischer et al., 2018), the LIG potentially provides insights into the drivers of sea level rise and the long-term impacts under a global temperature target set out in the 2016 Paris Climate Agreement (Schellnhuber et al., 2016).

Here we present version 1.0 of the Last Interglacial SST database (Turney et al., 2019). This database builds on the previously published 2010 data compilation of (Turney and Jones, 2010), and includes substantially more records. Importantly, the micro-organisms used to determine SSTs move along with the currents and encounter a range of temperatures during their life cycle (van Sebille et al., 2015;Doblin and van Sebille, 2016;von Gyldenfeldt et al., 2000). As a result, previous workers have suggested ocean drift of micro-organisms can have a major influence on reconstructed environmental change (van Sebille et al., 2015;Monroy et al., 2017;Kienast et al., 2016;Hellweger et al., 2016;Rembauville et al., 2016;Viebahn et al., 2016;Nooteboom et al.) and potentially explains the divergence between laboratory culture and core-top calibrations (Anand et al., 2003;Müller et al., 1998;Prahl et al., 2003;Sikes et al., 2005;Segev et al., 2016;Elderfield and Ganssen, 2000), and palaeoclimate estimates and model outputs (Otto-Bliesner et al., 2013;Bakker and Renssen, 2014;NEEM Community Members, 2013;Lunt et al., 2013), including the recently recognised historic (Anthropocene) change in modern plankton communities which has major implications for calibration studies (Jonkers et al., 2019). The influence of ocean currents has not been explored (or corrected for) in previous studies of the LIG (Hoffman et al., 2017;Capron et al., 2014;Turney and Jones, 2010) and is important for obtaining correct absolute SSTs. This descriptor describes the contents of the database, the criteria for inclusion, and quantifies the relation of each record with instrumental temperature, including the estimated impact of ocean current drift on individual sites and global averages. The current database includes a large number of metadata fields to facilitate the reuse of the data and identification of key records for future investigations into the LIG. Specific criteria were developed to gather all published proxy records that meet key objective and reproducible criteria. The database will be updated yearly as newly reported records are published.

## 2 Methods

### 2.1 Global Compilation

We have compiled a global network of published quantified SSTs using faunal and floral assemblages, Mg/Ca and Sr/Ca ratios of calcareous organisms, and $U^{K'}_{37}$ estimates across the period of record interpreted as representing the LIG. In many instances, we used the period represented by low $^{18}O$ values in benthic foraminifera shells (the lightest isotopic values during 90-150 kyr representing minimum global ice volume), although in some sequences, $\delta^{18}O$ values were reported and we relied on other complimentary proxies; for instance, the $CaCO_3$ content of sediments as a measure of glacial-interglacial variability (Turney and Jones, 2010;Cortese et al., 2013) (Figures 1 and 2). Whilst the age control points defining the plateaus in $\delta^{18}O$ and other proxies are not absolutely dated with chronological uncertainties of one to two millennia (Martinson et al., 1987;Lisiecki and Raymo, 2005), it is important to note that we are not aiming to resolve centennial and millennial-scale variability through the interglacial. We acknowledge that some individual SST estimates may not fall within the LIG or have been excluded (due to these chronological uncertainties) but we consider the averaging of values across the full interglacial provides a robust value for each record and ultimately the regional and global reconstructions.

We have therefore not attempted to generate a time series of sea surface temperatures through the LIG. Previous studies have highlighted that individual site $\delta^{18}O$ changes in benthic foraminifera (for instance, during deglaciation) may be offset by several millennia as a result of local deep-water temperature and $\delta^{18}O$ seawater variations (Govin et al., 2015;Waelbroeck et al., 2008) (Figure 2). In an attempt to bypass some of these issues, other studies have attempted alignment of marine records to speleothem-dated, ice core reconstructions (Hoffman et al., 2017) but modelled age uncertainties can be on the order of millennia (e.g. Hoffman et al. Fig. S7) while the assumed synchroneity of extra-regional changes has challenges; for instance, more than half of reported Pacific marine cores (those from the Northern Hemisphere) in a recent study were correlated to the Antarctic EPICA Dome C $\delta D$ (Hoffman et al., 2017), with warming in the south known to lead the north by one to two millennia (Hayes et al., 2014;NEEM Community Members, 2013;Kim, 1998;Rohling et al., 2019). The development of accurate and precise age estimates for LIG records is urgently needed to resolve the timing of global climate change but will require a considerable future international effort (Govin et al., 2015). Given the relatively large chronological uncertainties associated with comparing global SST time series (Hoffman et al., 2017;Govin et al., 2015;Capron et al., 2017) we have therefore not attempted to generate a time-series of changes within the LIG but instead determine average temperatures as a robust estimate of mean climatic conditions. Whilst not offering precisely-dated geochronological frameworks, the global ice minima as represented by the $\delta^{18}O$ plateau and/or associated proxy measures of interglacial conditions are sufficiently well-defined in all marine records to accommodate local deep-water temperature and $\delta^{18}O$ variations, sampling resolution and/or sedimentation rates to identify the LIG, thereby maximising the number of records that have reported quantified SSTs across the interglacial (Cortese et al., 2013;Govin et al., 2015); a minimum of three SST values across the LIG in each record were required for inclusion in our dataset. This is not to downplay the significance of millennial-scale climate variability across the LIG (Galaasen et al., 2014;Rohling et al., 2002;Tzedakis et al., 2018;Jones et al., 2017) but our approach does provide some benefits. Whilst our approach sacrifices temporal control, it does minimise the uncertainty on zonal and global temperature averages.

To quantify the temperature difference between the LIG and present day, we do not compare the LIG estimates to the relatively poor observational coverage of earlier periods, including the nineteenth century (pre-industrial) (Hoffman et al., 2017) or the long-term annual means calculated from 1900-1997 (Capron et al., 2014), both of which have considerable uncertainties given the limited network of 'observations' prior to the satellite era (Brohan et al., 2006;Huang et al., 2020). Here instead we report SSTs expressed as anomalies relative to global 'modern' instrumental and satellite observations across the period 1981-2010 obtained from HadISST (Rayner et al., 2003). Each LIG temperature record is linked to at least one literature source, the citation of which includes author(s), year of publication and typical archiving information (e.g. journal, volume, issue, pages, publisher and place of publication). Where multiple temperature estimates have been published over time from the same site, we chose the most recent publication for inclusion in the database (so long as the data were not flagged as erroneous) (Figure 3). Note that alkenone proxies are interpreted as providing annual SST estimates.

Here we use the mean temperature estimates to constrain the role of thermal expansion in global sea level rise across the LIG and provide boundary conditions for future modelling studies investigating the impact of warming on polar ice sheets. To determine the greatest possible contribution of warming to ocean thermal expansion and ice sheet melt, we used the published age models to identify the maximum annual SST within the first 5 kyr of the LIG (i.e. 129-124 kyr). For the purposes of this sensitivity analysis, the maximum temperatures were assumed to be synchronous globally, a scenario we recognise as unlikely but does provide an upper limit for warming in

the 'early' LIG. To provide an upper estimate on the magnitude of warming in polar waters over the deglaciation, we also report here the difference between late Marine Isotope Stage 6 mean SSTs (~140-135 kyr) and the maximum early LIG SSTs for ocean cores in the mid to high-latitudes. To calculate the anomaly relative to present day, we utilise SSTs from the nearest 0.5˚ latitude x 0.5˚ longitude averaged across the period 1981-2010 (Rayner et al., 2003). For the uncertainties calculated for the regional and global SST anomalies, we incorporate the uncertainties from the proxies (reported in the database), and the uncertainties associated with estimating regional and global temperatures from limited spatial coverage. To achieve this we propagated the SST uncertainties for each measurement through each of the averaging steps (i.e. temporal to grid cell to zonal to area-weighted global) in our ocean-area-weighted average (McKay et al., 2011). We used quoted uncertainty estimates for each study where reported; if not available, we applied proxy-specific uncertainty estimates. Although the impact of the spatial coverage was not explored in this study, it has been previously estimated using the same approach (McKay et al., 2011). In that study, the uncertainty associated with the limited spatial range of the oceanographic proxies was estimated by calculating 1000 random one-year global SST anomalies over the twentieth century, and compared to averages derived using only the palaeoceanographic network. No systematic biases were identified with a $1\sigma$ uncertainty estimated to be <0.1˚C. In this study, we have expanded the spatial network, and consider ±0.1˚C to be a reasonable, high-end estimate.

The database comprises six worksheets of data comprising maximum annual temperatures during the early LIG (defined here as the maximum temperature reported within the first five millennium of the LIG; 129-125 kyr), mean annual temperature, the Marine Isotope Stage 6/5 SST difference, December to February temperature (DJF; Northern Hemisphere winter and Southern Hemisphere summer), June to August temperature (JJA; Northern Hemisphere summer and Southern Hemisphere winter), and summary statistics (see Supplementary Information):

- The early maximum and mean annual SST dataset comprises 189 marine sediment and coral records from latitudes spanning from 55.55˚S (radiolaria assemblage transfer function reconstruction obtained from site V18-68) (CLIMAP, 1984) to 72.18˚N (planktonic foraminifera assemblage modern analogue technique from site V27-60) (Vogelsang et al., 2001)
- The mean December-February SST dataset comprises 99 marine sediment records from latitudes spanning from 61.24˚S (diatoms transfer function reconstruction obtained from site PS58/271-1) (Esper and Gersonde, 2014) to 72.18˚N (planktonic foraminifera assemblage modern analogue technique from site V27-60) (Vogelsang et al., 2001).
- The mean June-August SST dataset comprises 92 marine sediment records from latitudes spanning from 54.55˚S (radiolaria assemblage transfer function reconstruction obtained from site V18-68)(CLIMAP, 1984) to 72.18˚N (planktonic foraminifera assemblage modern analogue technique from site V27-60) (Vogelsang et al., 2001).

In total, the Last Interglacial SST database comprises a total of 203 unique sites described in 100 publications.

## 2.2 Ocean Drift

Crucially, modern calibration relationships are an average developed using a selected number of locations that will not necessarily capture the range of "signal drift". This drift is caused by the fact that planktic SST recorders can be transported over considerable distances in the water column before being deposited, which particularly applies to all those sites that lie under strong boundary currents or near major ocean fronts (van Sebille et al., 2015). Unfortunately, Ocean General Circulation Models (OGCMs) typically have insufficient spatial resolution to capture mesoscale features that are critical for modelling the lateral drift of particles (Nooteboom et al., 2020). To investigate the impact of drift on SST reconstructions, we therefore used contemporary ocean circulation as a first-order approximation for the LIG. Whilst we acknowledge that there was likely a weakening of the Atlantic Meridional Overturning Circulation (AMOC) during the early LIG (Shackleton et al., 2020;Turney et al., 2020;Thomas et al., 2020;Jones et al., 2017), subsequent recovery after 127 kyr appears to have established a global circulation comparable to present day as suggested by recent ocean $\delta^{13}$C modelling results across the mid-interglacial (Bengtson et al., 2020). We performed an experiment with virtual particles in an eddy-resolving ocean model (the Japanese Ocean model For the Earth Simulator or OFES) (Masumoto et al., 2004), which has a 1/10° horizontal resolution and near-global coverage between 75°S and 75°N (van Sebille et al., 2012). Utilising the 3D velocity field of the model, we used the Parcels code (oceanparcels.org) (Lange and van Sebille, 2017) to compute the trajectories of more than 170,000 virtual planktic particles that end up at each of the sites by tracking them backwards in time, first simulating the sinking to these sites at 200 m/day and subsequently the advection at 30 m depth for a lifespan of 30 days; coral SSTs were not corrected for drift. Given the lifespan of most organisms that have been used to generate a temperature signal (Jonkers et al., 2015;Bijma et al., 1990), we consider a 30-day drift provides a reasonable estimate of the

drift distance. Previous work has demonstrated comparable uncertainties between different models (van Sebille
et al., 2015), providing confidence in the use of the OFES for the purposes of this study.
During the 30-day lifespans, we recorded the temperatures along the trajectories and compared those to the local
temperature at 30 m water depth at the site where the particles would end up on the ocean floor. This resulted in
daily temperature anomalies along the trajectories, which were averaged through the lifespan and over the 840
virtual particles that ended up at each site, and then subtracted from the reported LIG estimates (Figure 1 and
Database). With the recent recognition that core-top calibrations may be incorrect given historic changes in
marine communities that have accompanied anthropogenic warming (Jonkers et al., 2019), it should be noted
that SST proxy calibrations based on regional core-top calibrations may give an incorrect absolute value, an
aspect that will form the focus of future work.

**2.3 Hemispheric and Global Calculations**

Global mean SST anomalies were calculated by averaging anomalies in a 10° latitude × 10° longitude grid, then
averaged globally after weighting for the area of ocean in each grid cell (Figure 5). The uncertainty calculated
for global SST anomalies incorporates uncertainties in the SST proxies as reported in the original studies, which
typically ranges from 1 to 2°C, and is then propagated through subsequent steps in the analysis. Additional
uncertainty associated with estimating global anomalies from limited spatial coverage, and the potential impacts
of age uncertainty or averaging non-synchronous data are not considered here. Consequently, the derived
estimates do not capture all of uncertainty in global and zonal SST anomalies, however, the zonal consistency of
the results suggest that the signal is large enough to overcome these unquantified sources of uncertainty.
Furthermore, whilst some regions may exhibit substantial differences arising from drift (Figure 4), taken
globally the mean annual temperature estimates are comparable (Figure 5). The new LIG SST dataset allows us
to report the estimated thermosteric contribution for LIG sea levels using the method reported by (McKay et al.,
2011). We use the above temperature changes to calculate the thermosteric contribution to LIG sea levels by
using the Thermodynamic Equation of Seawater 2010 (TEOS-10). To provide an estimate of thermosteric sea
level rise, we explored a range of scenarios where warming penetrated different ocean depths: 700 m, 2000 m
(approximately the upper half of the ocean) and 3500 m (the whole ocean). We determined the change in the
specific volume of the warmed water column of each a 10° latitude × 10° grid cell while holding the salinity
constant and neglecting changes in ocean area. Here absolute temperature is considered, as specific volume is
more sensitive to temperature changes at warmer temperatures.

**3 Results and Discussions**

**3.1 Quality Control**

The Last Interglacial SST database is derived from published articles that have already been peer-reviewed. To
generate the database, we undertook a comprehensive check to remove duplicate records, erroneous location
information and other errors. In addition to ensuring consistency of data processing and any recalculations (for
instance, sea-surface temperature anomalies relative to the period CE 1981-2010), we also checked uncertain
metadata reported for individual sites, and directly communicated with selected article authors and/or other
experts as part of the record-validation process.

**3.2 Ocean Circulation**

A challenge for the Last Interglacial is determining what influence (if any) ocean circulation had on the
temperatures experienced (and reconstructed) by organisms that are used to generate SST reconstructions.
Addressing this issue is an important objective of the current study but we found the magnitude of temperature
offset (bias) is limited to only a few key locations (Fig. 1), with similar final reconstructions for individual sites,
latitudinally-averaged and globally average temperatures (Figures 4 and 5, and Table 1). This provides an
important check of our temperature recalculations. As a sensitivity test, we therefore explored virtual planktic
particles that 'live' for 30 days to investigate whether a prolonged period of drift made a discernible difference
(data not reported here). Only a few species have been suggested as living for a longer period of time. For
instance, in laboratory experiments the planktic foraminifer *Neogloboquadrina pachyderma* sinistral has been
shown to survive up to 230 days (Spindler, 1996) but this species may be an exception due to its ability to
survive in sea ice (Dieckmann et al., 1991).
Using 30-days' drift to simulate the travelling time/lifespans of virtual planktic particles in the upper part of the
water column, we quantified the inherited temperature signal of flora/fauna at each site in the database. The
virtual microorganisms with a 30-day 'lifespan' travelled from a few tens to a few hundreds of kilometres. The

temperature offsets are almost all positive in the tropical East Pacific, the North Atlantic and South China Sea,
meaning that the planktic particles originated from warmer climates and hence record a higher temperature
estimate than local conditions would suggest; with the opposite effect observed in the western tropical Pacific
and Southern Ocean (Figure 1). The offset can be substantial – with values ranging from -6.9°C for site MD98-
2162 at 4.7°S in the tropical West Pacific (Visser et al., 2003) and up to 3.5°C in site RC13-110 on the Equator
(Pisias and Mix, 1997) – with the largest changes associated with boundary currents and major ocean fronts.
Intriguingly, these values are comparable to the difference previously reported for Mg/Ca foraminifera core-top
calibration with those obtained from laboratory-cultured Mg/Ca calibrations (Elderfield and Ganssen,
2000;Hönisch et al., 2013). Both the uncorrected and 30-day drift temperatures are provided in the database.
These temperature reconstructions led to statistically indistinguishable global temperature (and thermosteric sea
level change; Figure 5). Users of the database are therefore able to use either the authors' original sea-surface
temperature determinations or our drift-corrected estimates, as required.
**3.3 Proxy and Seasonal Effects**
To evaluate potential biases in our analysis, we further subsampled our database by proxy type (Figure 4). The
large network of sites and proxies do not appear to demonstrate any significant offset in annual reconstructions
(at least within the uncertainty of the reconstructions), although there is a tendency for alkenone temperatures to
be at the upper end of the range, implying there may be a seasonal bias, as reported previously (Hoffman et al.,
2017). Importantly, we also compiled seasonal quantified temperature estimates that have been reported as the
seasonal warmest or coolest months in the year (taken here to represent June-August and December-February
depending on the hemisphere being considered). Our result suggests that any bias, if real, is smaller than the
uncertainties at the global or zonal level reported here. Intriguingly, the warmest month estimates for the high
latitudes in both hemispheres have more muted warming than the mean annual estimates while the low to mid
latitudes exhibit considerably cooler estimates (Table 1). In contrast to the alkenone estimates for the annual
estimates, the more muted response of foraminifera, radiolaria and diatoms for the seasonal reconstructions
implies they are influenced by a larger part of the seasonal cycle. We therefore consider that seasonal
reconstructions should be treated as conservative estimates of temperature for the LIG.
**3.4 Average and Early Temperatures during the Last Interglacial**
We find global average annual temperatures across the full duration of the LIG were only marginally warmer
than present day. We derive a global mean annual temperature anomaly of $0.2 \pm 0.1$°C, the same value obtained
after correcting for drift (Table 1). These values, however, mask considerable zonal differences, with
significantly cooler mean annual uncorrected temperatures (i.e. not corrected for drift) within 23.5° of the
equator ($-0.3 \pm 0.2$°C) and amplified warming polewards (Figure 5). Ideally, we would have a dense network of
records in the mid- to high-latitudes for investigating the impact of warming surrounding polar ice sheets but
unfortunately the number of sites and their spatial distribution do appear to have an impact on the reconstructed
values. Comparison of the SST anomalies poleward of 45° and 50° latitude (Table 1) shows substantial
differences, most notably in the Southern Hemisphere where a large increase in zonally averaged SST occurs
alongside a decrease in the number of records polewards of 50°S (Table 1). For instance, the drift-corrected
SSTs for the LIG are $0.8 \pm 0.3$°C (n=13) and $2.7 \pm 1.1$°C (n=3) polewards of 45°S and 50°S respectively. It
should also be noted that whilst the Northern Hemisphere polar estimates are similar for both latitudinal ranges,
the majority of sites are in the North Atlantic, with limited representation in the Pacific Ocean (Figure 1). We
therefore recommend that when considering mid- to high-latitude zonal SST averages, the values derived from
records polewards of 45° are more likely robust but acknowledge these may be conservative estimates (with
considerably larger warming further to the south). We therefore estimate uncorrected 'polar' warming in the
Northern Hemisphere to be $2.0 \pm 0.4$°C, and in the Southern Hemisphere, $0.2 \pm 0.3$°C (Table 1). Correcting for
drift decreased the northern estimate to $1.5 \pm 0.4$°C and increased in the south to a mean annual SST to $0.8 \pm 0.3$
332 °C.
The maximum temperatures of the early LIG were up to $0.9 \pm 0.1$°C warmer than 1981-2010, regardless of
whether the values were corrected for drift (Table 1 and Figure 6). Similar to the mean SSTs of the LIG, there
appears to have been considerable zonal differences in the uncorrected values: $0.1 \pm 0.2$°C within 23.5° of the
equator, $3.2 \pm 0.4$°C polewards of 45°N, and $1.5 \pm 1.1$°C polewards of 45°S. After correcting for drift, the
estimated SST in the north changed to $2.8 \pm 0.4$°C and in the south, to $2.1 \pm 1.1$°C. The latter estimate from the
Southern Hemisphere is ~2°C (relative to 1981-2010), potentially providing an important constraint for future
Antarctic ice-sheet model simulations for the LIG (Turney et al., 2020;Golledge et al., 2015). These data
support previous work which have reported substantial polar temperature amplification during the LIG,
particularly in the Northern Hemisphere (described in the literature as 'Arctic amplification') (Overpeck et al.,
2006;Mercer, 1978;Mercer and Emiliani, 1970;Thomas et al., 2020;Miller et al., 2010). The global temperature
pattern closely follows insolation changes across this period, during which the Earth's greater eccentricity led to

reduced radiation over the equator and more intense high latitude spring-summer insolation (Figure 2)
(Overpeck et al., 2006;Hoffman et al., 2017). Comparison to Marine Isotope Stage 6 SSTs appears to show the
greatest warming in the northeast Atlantic and south Atlantic (Figure 7), suggesting Greenland and the West
Antarctic ice sheets would have been particularly vulnerable to warming in the early interglacial (Clark et al.,
2020;Turney et al., 2020;Dutton et al., 2015;Mercer, 1978) though we cannot resolve the relative timing of mass
loss in this analysis (Rohling et al., 2019;Hayes et al., 2014). Recent work suggests the earliest warming took
place in the Atlantic (and Indian) Ocean sectors of the Southern Ocean (Chadwick et al., 2020), consistent with
our findings. However, our observed polar warming is larger than some climate model simulations, implying the
latter are failing to capture one or more key feedbacks (e.g. carbon, sea-ice and ice-sheet feedbacks) in the
climate system (Bakker et al., 2013;Otto-Bliesner et al., 2013;Thomas et al., 2020;Clark et al., 2020;Fogwill et
al., 2015).
**3.5 Thermal Expansion Contribution to Last Interglacial Sea Level**
The LIG is characterised by higher GMSL than present day (+6.6 to +11.4 m) (Grant et al., 2014;Dutton et al.,
2015;Turney and Jones, 2010;Rohling et al., 2017;Rohling et al., 2019). Here we quantified the contribution of
the relatively high temperatures on global sea levels through ocean thermal expansion for warming down to
2000 m ocean depth (Table 2). We find that through the LIG, the average SSTs contribution to thermosteric sea
level was negligible, approximately $0.05 \pm 0.10$ m uncorrected for ocean drift and $0.08 \pm 0.10$ m corrected for
drift, consistent with a recent reconstruction of near-modern global ocean heat content and negligible
thermosteric sea level rise (Shackleton et al., 2020). But for the early LIG (129-124 kyr), using our maximum
SST estimate, we obtained high-end estimate of thermal expansion to GMSL of $0.36 \pm 0.10$ m (uncorrected) and
$0.39 \pm 0.10$ m (drift corrected). These quantified estimates are comparable to a previously reported value of 0.4
$\pm 0.3$ m (McKay et al., 2011) which used the same methodology as here but a smaller network of SST records.
However, we should recognise that the depth of ocean warming is uncertain, and could have extended deeper
than 2000 m. If we assume warming penetrated the full ocean depth (down to 3500 m), we obtained a maximum
early LIG thermosteric sea level rise of $0.67 \pm 0.10$ m (uncorrected) and $0.72 \pm 0.10$ m (drift corrected) (Table
2). The recently reported early LIG (~129 ka) peak in global ocean heat content reconstructed from isotopic
ratios of atmospheric trace gases has determined a maximum thermal expansion of $0.7\pm0.3$ m (Shackleton et al.,
2020). To achieve ~0.7 m of thermosteric sea level rise during the early interglacial peak in temperatures, we
have to use both our maximum estimate of temperature rise, and our maximum estimate of the depth of
warming. A a recent modelling-proxy estimate proposed a range of 0.08 to 0.51 m for peak LIG warmth centred
on 125 kyr (Hoffman et al., 2017), which is more consistent with our results. Even though 125 ka is later than
the peak in global ocean heat content, this is effectively the same event but represents the age uncertainties in
the marine records. Although some uncertainty remains in the amplitude of thermal expansion between these
studies, it is clear that the sustained high global sea levels across the LIG and the limited role of warming on
thermal expansion implies a greater contribution from ice sheets, mountain glaciers, permafrost and
hydrological change. With the greatest warming relative to Marine Isotope Stage 6 in Atlantic basin (Figure 7),
our results are consistent with previous studies suggesting substantial mass loss from Greenland and the West
Antarctic Ice Sheet early in the Last Interglacial (Clark et al., 2020;Turney et al., 2020;Dutton et al.,
2015;Mercer, 1978;Hayes et al., 2014;Rohling et al., 2019).

**4 Data Availability**

The Last Interglacial SST database is provided as an Excel workbook in Supplementary Information and on the
PANGAEA Data Publisher at https://doi.pangaea.de/10.1594/PANGAEA.904381 (Turney et al., 2019); the data
is also available on the NCEI-Paleo/World Data Service for Paleoclimatology at
https://www.ncdc.noaa.gov/paleo/study/26851. This release comprises a single Excel file, tab delimited. We
welcome contributions from authors of additional or clarifying information. These will be incorporated into any
subsequent iteration of the database. When using data in this compilation, the original data collector(s) as well
as the data compiler(s) will be credited. Given the typically large uncertainties in the absolute dating of each
individual record, no attempt has been made to develop individual time series, and only mean values across the
Last Interglacial have been compiled. For simplicity we record the 1σ (68%) confidence interval in the site
temperature reconstructions. The inclusion of key metadata allows users to interrogate individual records for
their own appropriate screening criteria.

## 5 Conclusions

During the Last Interglacial (LIG; 129-116 kyr), global temperatures were up to 2˚C warmer than present day with marked polar amplification and global sea levels between 6.6 and 11.4 m higher than present day, offering a powerful opportunity to obtain key insights into the drivers of future change (a so-called 'process analogue'). The contributions of different sources to the LIG sea level highstand remain highly uncertain, however. As a result of relatively warmer surface temperatures, ocean thermal expansion has previously been estimated to have contributed $0.4 \pm 0.3$ m. To more precisely constrain this contribution to global mean sea level we report a new comprehensive database of quantified SSTs estimates derived from faunal and floral assemblages, Mg/Ca and Sr/Ca ratios of calcareous organisms, and $U^{K'}_{37}$ estimates from records spanning 55.55˚S to 72.18˚N. Here we have calculated maximum annual SSTs during the early interglacial (129-124 kyr) and mean annual SSTs through the LIG (129-116 kyr; 189 sites) alongside mean December-February (99 records) and June-August (92 records) values. Temperatures are reported as anomalies relative to the period CE 1981-2010. To estimate the temperature footprint arising from ocean circulation we also report SST anomalies corrected for 30-day drift, to simulate the travelling time/lifespans of virtual planktic particles in the upper part of the water column. Our reconstruction suggests an early LIG maximum global mean annual SST of $0.9 \pm 0.1$˚C and an average warming across the LIG of $0.2 \pm 0.1$˚C. However, these values are strongly driven by polar warming of several degrees, with little to no warming in the tropics. We find the influence of warming on ocean thermal expansion to have had a limited influence on global mean sea levels across the full LIG, but with a likely range of between $0.39\pm0.1$ m and $0.72\pm0.10$ m early in the interglacial. Our findings therefore imply a relatively greater contribution of ice sheets, mountain glaciers, permafrost and hydrological change to global sea level during the LIG, likely driven by polar amplification of temperatures. We hope this database may provide a springboard for future studies that can bring to bear new geochronological methods (e.g. tephra) to constrain the age models of individual sequences to sub-millennial uncertainty, something currently not possible for most reported marine sequences. An improved network of high-resolution, well-dated and quantified LIG climate reconstructions (particularly in data-sparse locations) will enable precise integration of ice sheet, marine and terrestrial records to better understand Earth system responses to high-latitude warming. The Southern Ocean and North Pacific are regions where major knowledge gaps currently exist.

**Supplement.** The supplementary figures and version 1.0 of the database (Excel file) can be accessed via the *Earth System Science Data* discussion page of this manuscript.

**Author contributions.** RTJ and CSMT conceived the research; CT, NPM, EvS, and ZT designed the methods and performed the analysis; CT wrote the paper with substantial input from all authors.

**Competing interests.** The authors declare that they have no conflict of interest.

**Acknowledgements.** It is with great sadness that our close friend and colleague Richard T. Jones was not alive to see the publication of this study. Without Richard this work would not have been possible. He is sorely missed. CSMT and CJF were supported by their Australian Research Council (ARC) fellowships (FL100100195 and FT120100004). We would also like to acknowledge the important role of the International Ocean Discovery Program (IODP), the Australian and New Zealand International Ocean Discovery Consortium (ANZIC), and the previous scientific ocean drilling programs, the results from which underpin this study and without whom this analysis would not have been possible. We are grateful to the four reviewers and editor for helping improve the first draft of this manuscript.

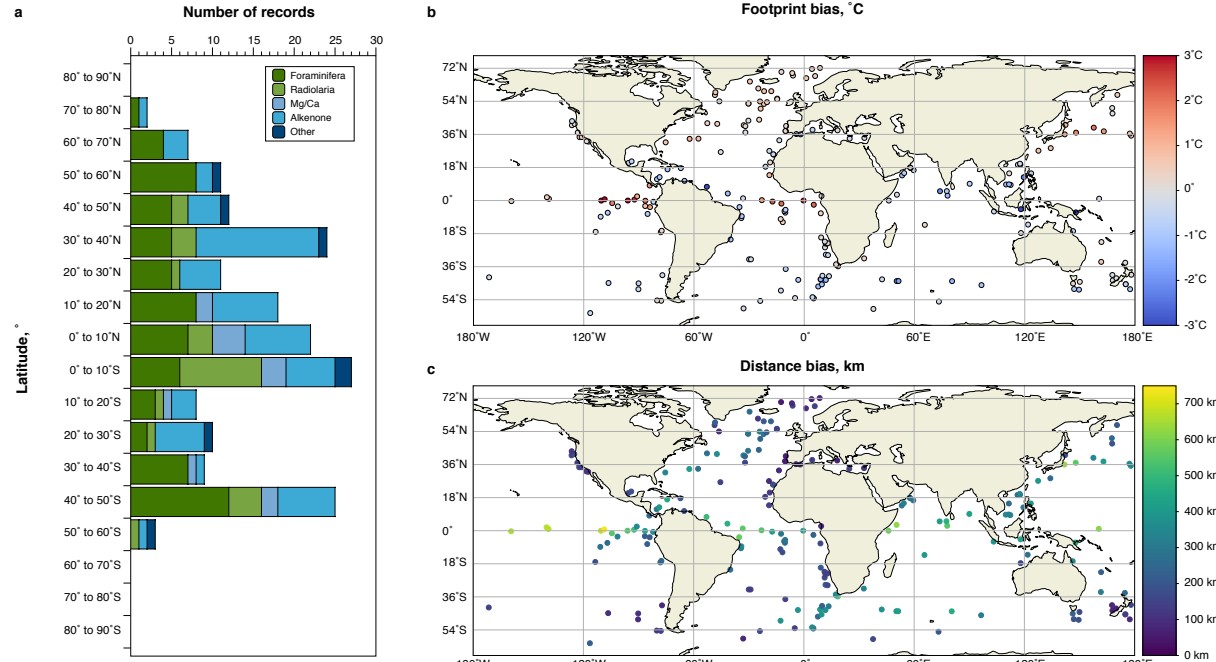

**Figure 1: Last Interglacial proxy-based annual sea surface temperature dataset and modelled inherited signal.**
Histogram showing the number of Last Interglacial records of annual sea surface temperature binned by 10° latitude (panel
a) with virtual microfossil temperature offsets defined as the difference between along-trajectory recorded temperatures and
local temperatures (panel b) and distance (panel c) travelled in the Japanese Ocean model For the Earth Simulator (OFES;
run between CE 1981 and 2010) determined for 30-day 'lifespans'(van Sebille et al., 2015).

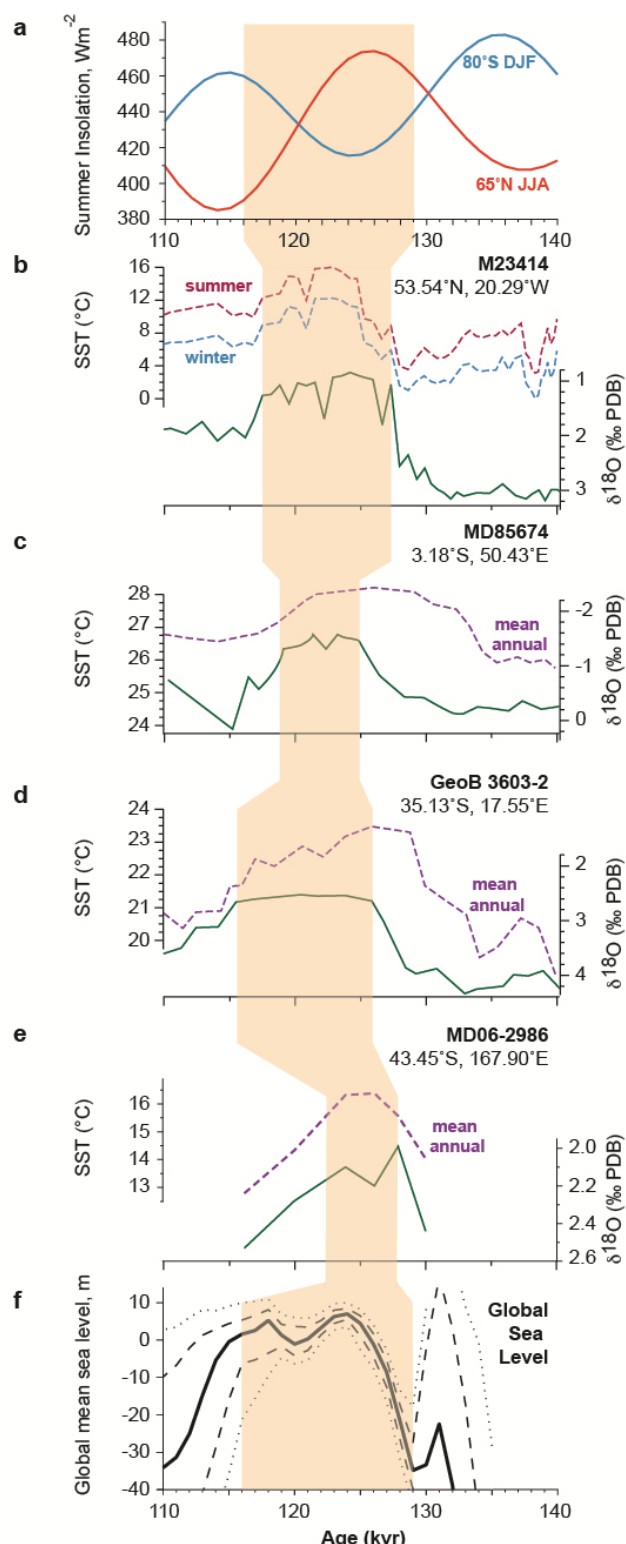

**Figure 2: Relationships between δ¹⁸O plateau and sea surface temperatures and environmental changes across the**
**Last Interglacial.** (a) Insolation changes calculated from ref. (Laskar et al., 2004). Sea surface temperatures (dashed purple
lines) across the Last Interglacial (light orange shading) compared to the benthic foraminifera δ¹⁸O (solid green lines) for
selected sites in different ocean basins: (b) M23414 (North Atlantic) (Kandiano et al., 2004), (c) MD85674 (equatorial
Indian Ocean) (Bard et al., 1997), (d) GeoB 3603-2 (southern Indian Ocean) (Schneider et al., 1999), and (e) MD06-2986
(southern Pacific Ocean) (Cortese et al., 2013). (f) The probabilistic reconstructed global sea level curve is reported by
(Kopp et al., 2009); heavy lines mark median projections, dashed lines the 16th and 84th percentiles, and dotted lines the
2.5th and 97.5th percentiles.

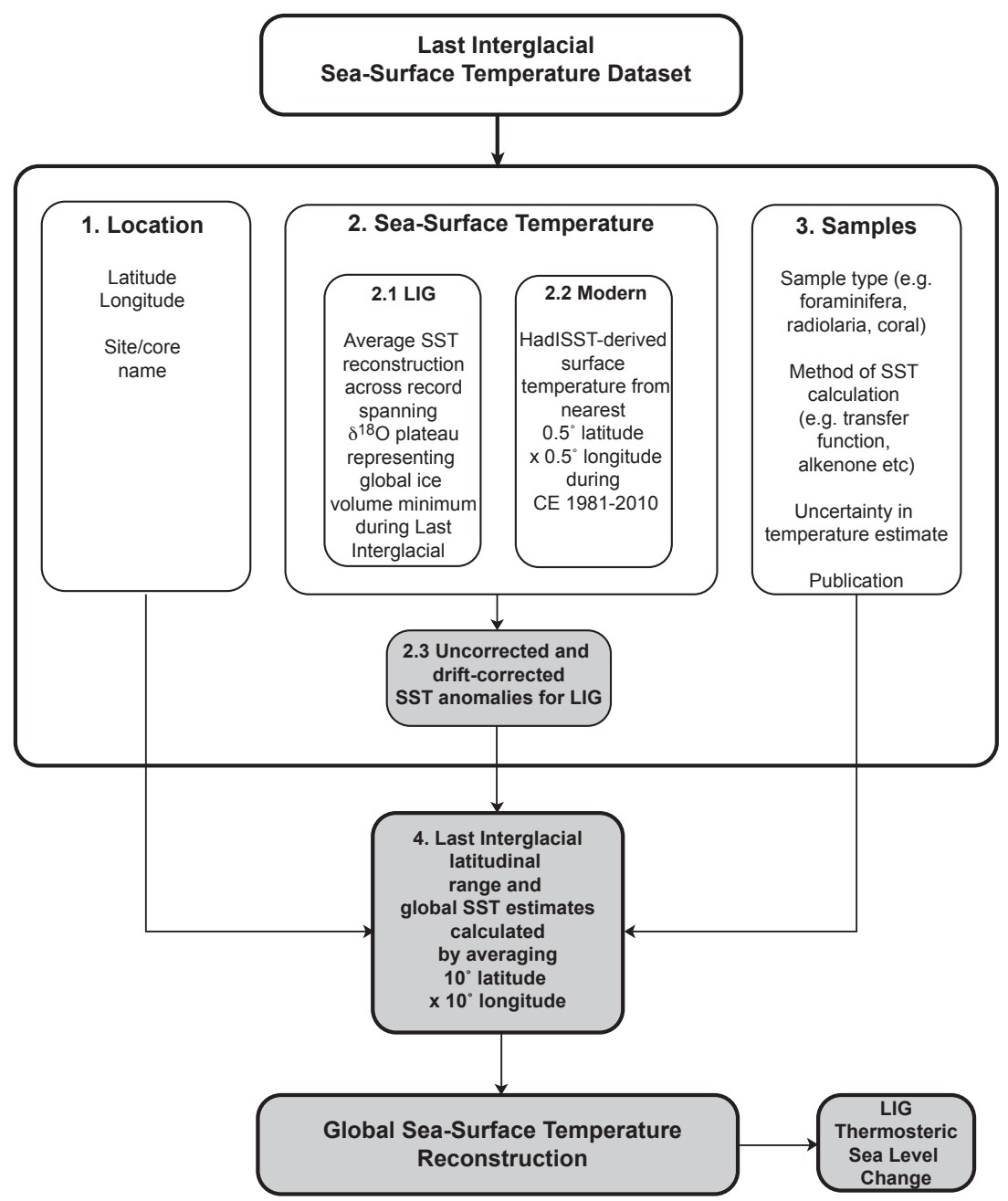

**Figure 3: Simplified scheme for the generation of the Last Interglacial sea-surface temperature database providing an overview of the data collection and processing.** The numbered boxes set out the stages required to generate a global database of surface temperatures from marine records: 1. Location; 2. Last Interglacial and modern SSTs (including drift calculation); and 3. Metadata including method of temperature reconstruction and associated uncertainty. Grey boxes indicate additional processing of data from the original publications, generating new outputs (which are provided in the database).

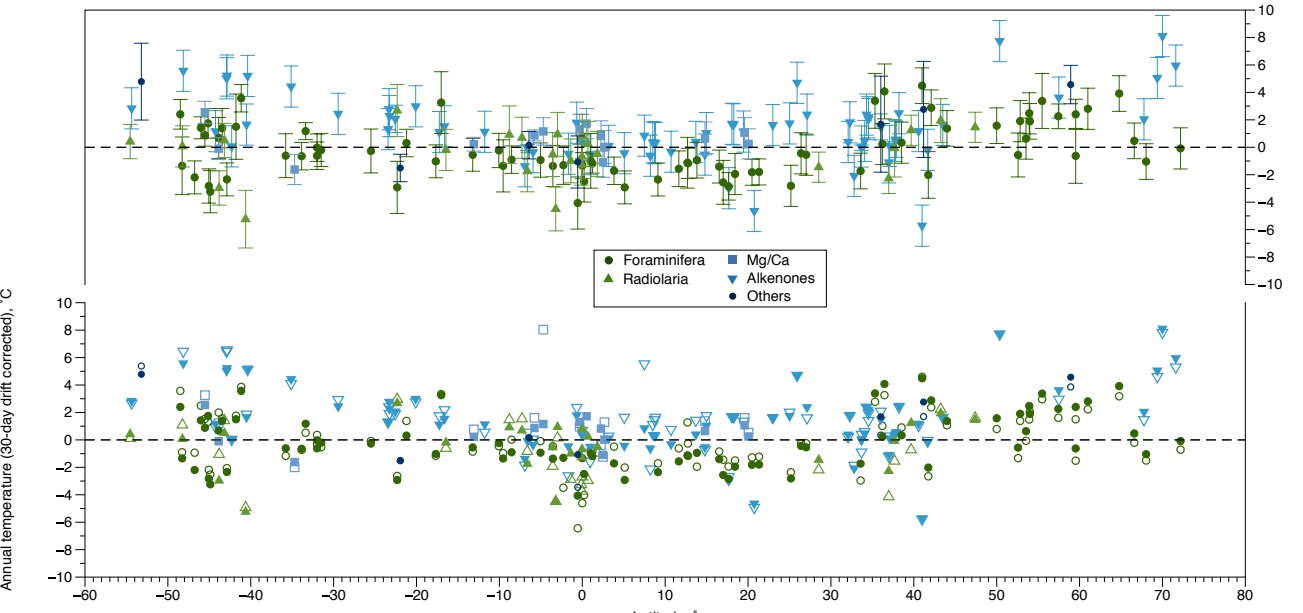

**Figure 4: Quality-control plot of latitudinal distribution of proxy mean annual Last Interglacial sea-surface**
**temperature anomalies.** Estimates given relative to the modern period (1981-2010) (Rayner et al., 2003) with no drift
correction (upper panel) and 30-days drift (lower panel). Lower panel shows drift-corrected SSTs as open symbols with the
uncorrected SSTs given as filled symbols. Uncertainties on upper panel given at 1σ.

**Sea Surface Temperature**

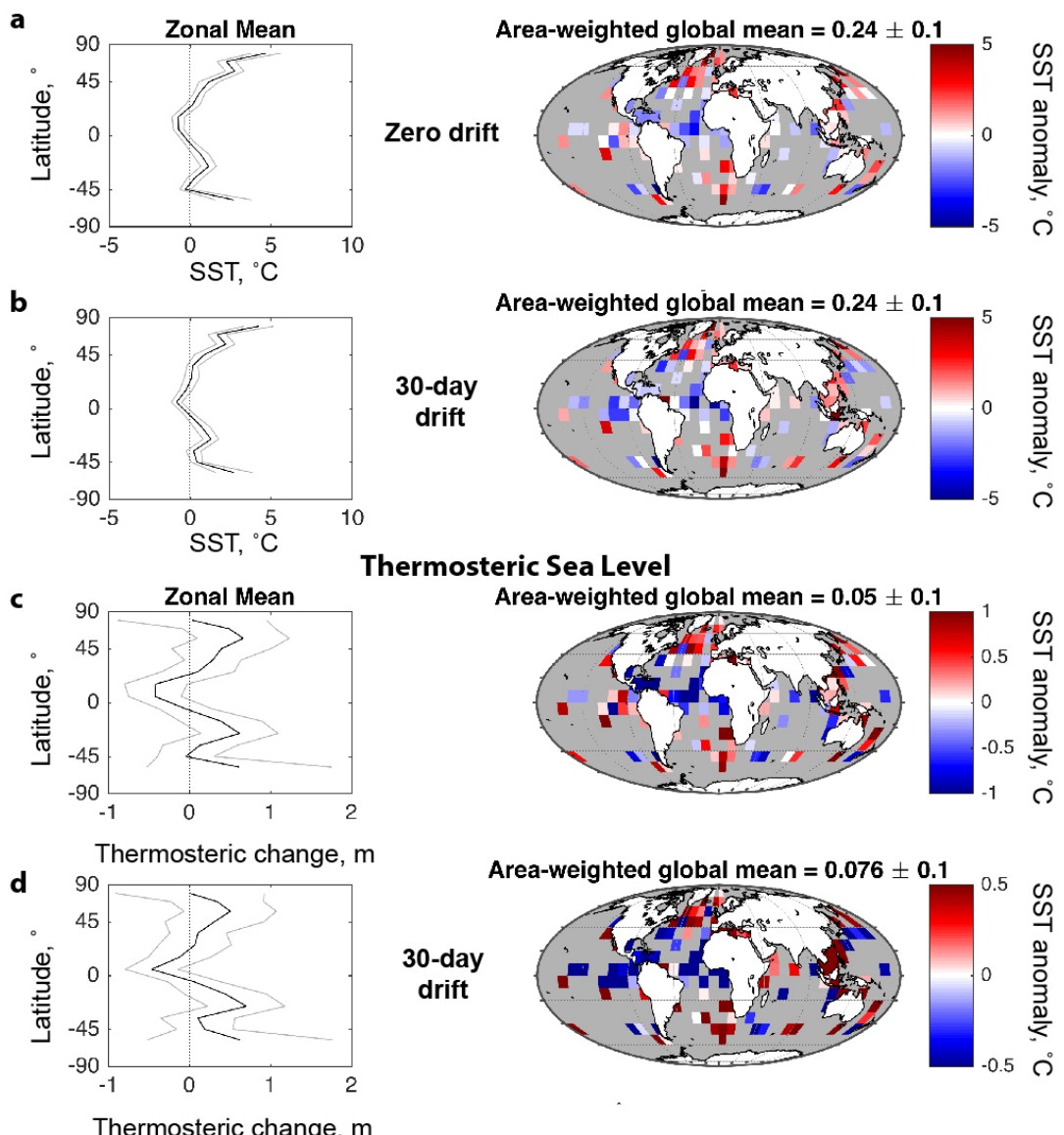

**Figure 5: Global and zonal mean annual sea-surface temperature (SST) anomalies and thermosteric sea level change across the full Last Interglacial**. Temperature anomalies reported as uncorrected (panels a and c respectively) and after applying 30-day (panels b and d respectively) temperature offsets arising from ocean current drift. Uncertainty for zonal average reconstructions given at 1σ. Here ocean warming is assumed to have penetrated to 2000 m depth, on average. Temperature estimates relative to the modern period (CE 1981-2010).

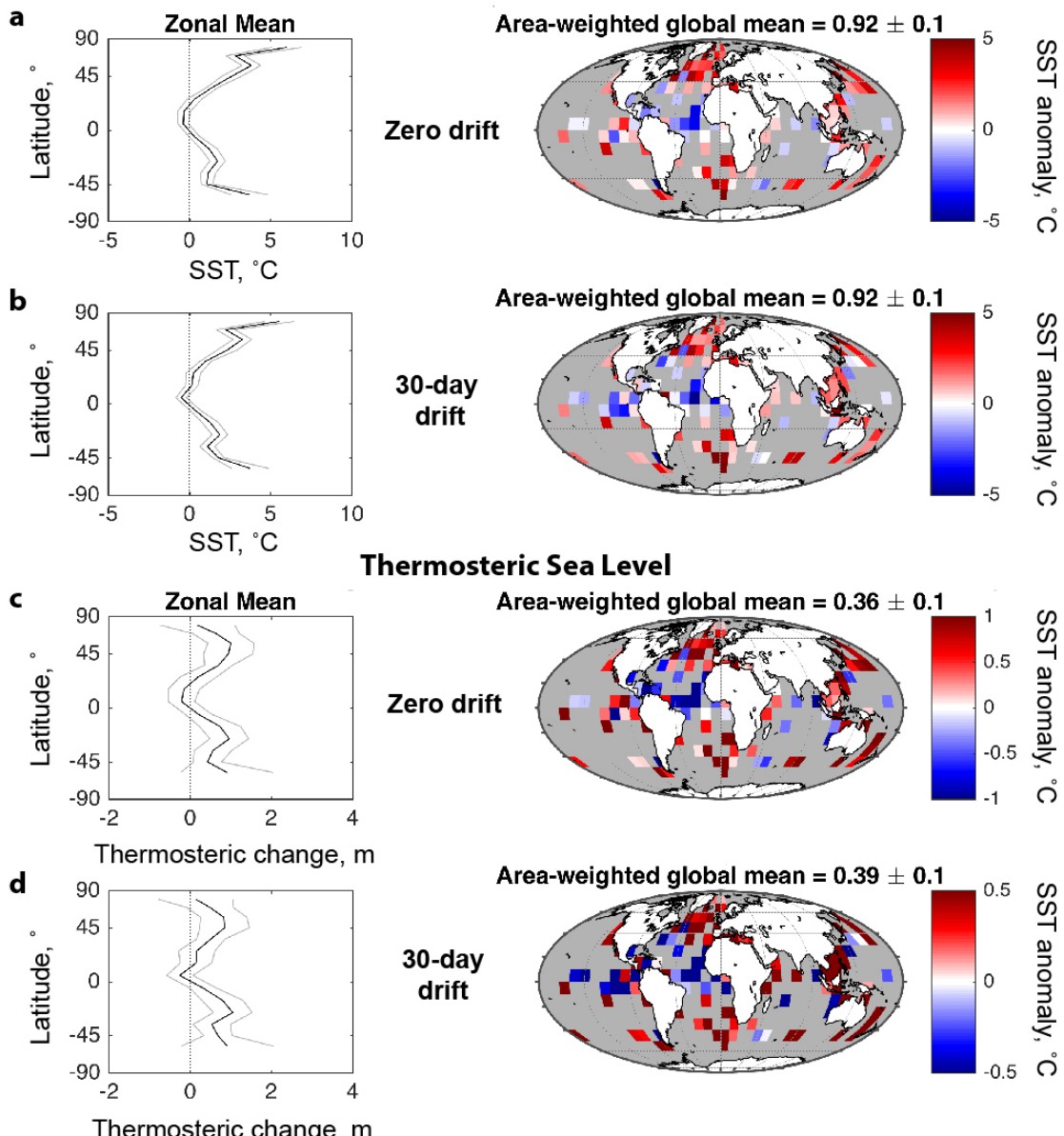

**Figure 6: Global and zonal mean annual sea-surface temperature (SST) anomalies and thermosteric sea level change during the early Last Interglacial**. Temperature anomalies reported as uncorrected (panels a and c respectively) and after applying 30-day (panels b and d respectively) temperature offsets arising from ocean current drift. Uncertainty for zonal average reconstructions given at 1σ. Here ocean warming is assumed to have penetrated to 2000 m depth, on average. Temperature estimates relative to the modern period (CE 1981-2010).

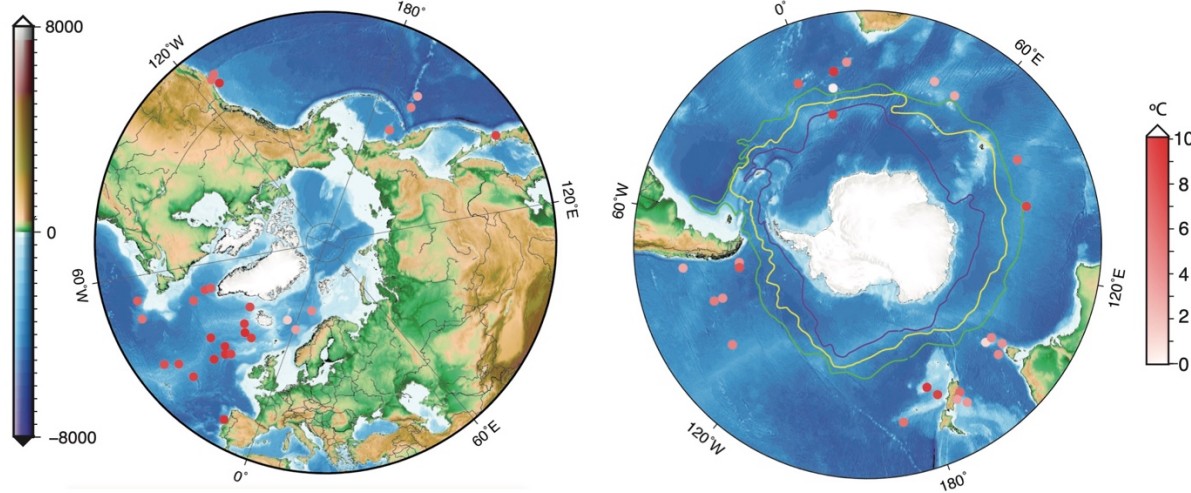

**Figure 7: Mid- to high-latitude sea surface temperature (SST) difference between late**
**Marine Isotope Stage 6 and maximum values of the early Last Interglacial (Stage 5).**
Map made using Generic Mapping Tools (GMT) (Wessel et al., 2013).

|  | Global SST (°C) | Tropical SST (23.5°N to 23.5°S) | SST Polewards of 45°N | SST Polewards of 50°N | SST Polewards of 45° S | SST Polewards of 50°S |
|---|---|---|---|---|---|---|
| **Maximum Early LIG** | | | | | | |
| **(n)** | (189) | (87) | (22) | (20) | (13) | (3) |
| *Uncorrected* | 0.9 | 0.1 | 3.2 | 3.8 | 1.5 | 3.7 |
| *30-day drift* | 0.9 | 0.1 | 2.8 | 3.2 | 2.1 | 3.7 |
| *1σ* | 0.1 | 0.2 | 0.4 | 0.4 | 0.3 | 1.1 |
| **Mean (n)** | (189) | (87) | (22) | (20) | (13) | (3) |
| *Uncorrected* | 0.2 | -0.3 | 2.0 | 2.8 | 0.2 | 2.7 |
| *30-day drift* | 0.2 | -0.3 | 1.5 | 2.3 | 0.8 | 2.7 |
| *1σ* | 0.1 | 0.2 | 0.4 | 0.4 | 0.3 | 1.1 |
| **DJF (n)** | (99) | (35) | (16) | (15) | (14) | (9) |
| *Uncorrected* | -0.6 | -0.7 | -0.1 | 0.0 | -0.3 | 0.8 |
| *30-day drift* | -0.7 | -0.9 | -0.5 | -0.7 | 0.3 | 1.0 |
| *1σ* | 0.2 | 0.3 | 0.4 | 0.5 | 0.3 | 0.3 |
| **JJA (n)** | (92) | (35) | (20) | (19) | (4) | (1) |
| *Uncorrected* | -0.4 | -1.1 | 1.3 | 1.3 | -1.9 | 0.1 |
| *30-day drift* | -0.5 | -1.2 | 0.9 | 0.7 | -1.2 | -0.2 |
| *1σ* | 0.2 | 0.3 | 0.4 | 0.4 | 0.4 | 1.1 |

**Table 1: Annual and seasonal temperature estimates for the Last Interglacial.** DJF: December to February; JJA: June to August. Temperature anomalies relative to the period CE 1981-2010. Maximum early temperature is defined as the maximum annual temperature recorded during the estimated first five millennia of the Last Interglacial.

|  | Global sea level (m) | | |
| --- | --- | --- | --- |
|  | 700 m depth | 2000 m depth | 3500 m depth |
| **Maximum Early LIG (n=189)** | | | |
| *Uncorrected* | 0.12 | 0.36 | 0.67 |
| *30-day drift* | 0.13 | 0.39 | 0.72 |
| *1σ* | 0.10 | 0.10 | 0.10 |
| **Mean (n=189)** | | | |
| *Uncorrected* | 0.00 | 0.05 | 0.10 |
| *30-day drift* | 0.01 | 0.08 | 0.15 |
| *1σ* | 0.10 | 0.10 | 0.10 |

**Table 2: Annual temperature contributions to sea level during the Last Interglacial for different warming depths**.

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
