# Peer review of "A global mean sea-surface temperature dataset for the Last 1"

_Earth System Science Data, 2019_

## Short Comment (SC1) · 12 Feb 2020

This is a welcome work that tackles a key question that is presently still insufficiently resolved: understanding global and regional temperatures during a key instance of past warm climate. It is ideal that independent groups of researchers address the same problem with different approaches and producing comparable results, something that also addresses the hotly discussed issue of reproducibility in the sciences at large. This study parallels a number of previous efforts, and most closely the recent work of Hoffman et al (2017). The main differences with that study are, in subjective order of

importance: ocean drift correction is applied; SSTs are integrated across the whole LIG; a larger sample of SST proxy records; much larger sampling of seasonal SSTs.

The accounting of the oceanographic footprint of the proxy records seems to me the clearest novelty introduced in this work. This is very timely, and the importance of the drift is clear as seen in the biases in Fig. 1, although I expected this to also impact the global SST estimate. The authors provide some sensitivity test on the choice of the lifespan parameter of the virtual particles, but I find this aspect somewhat incomplete, as it focused only on parameters appropriate for foraminifera. In a sensitivity test, only lifespans longer than the 30-day value adopted in the database are tested, while shorter lifespans seem plausible for coccolith-based reconstructions, which make up much of the database; the sinking speed of 200 m/day and the 30 m depth for the lifecycle may not be adequate to simulate the situation with coccoliths and other organisms smaller than foraminifera, and with phytoplankton that is confined to the photic zone. I am not expert in these organisms, but it should have been relatively easy to apply different parameters to the main type of organisms relevant to the database (that is, if the literature suggests that these are substantially different from those used), and at least test the effect of taking unique values for the whole database when a differentiation could have been possible. Also, while this probably exceeds the scopes of this study, would it be possible to mention why a simulation of OFES with LIG boundary conditions is not contemplated, e.g., initiated with data from the coarser grid of an ocean model from a PMIP4 GCM? Maybe an idea for future work.

The integration of SSTs across the whole period has both advantages and pitfalls: on the one hand it makes results independent from the delicate set of choices that necessarily come with assessing age models and aligning them within and across basins on a coherent chronology; on the other hand it dismisses the millennial scale variability that is critical to understand notable climatic variability within the LIG. The authors recognize this, but I suggest that a more convincing explanation could be provided of the choice of working from the hypothesis (as in Turney and Jones 2010) of global syn-

chronicity of peak SSTs: why is it superior to other solutions that make some use of the each record's explicit age models, what are the implications of the assumption for the results?

Last, it is important that the results are discussed in the light of the new results on mean LIG ocean temperature based on Antarctic noble gas, in the paper by Shackleton et al. just out in January (2020; doi: 10.1038/s41561-019-0498-0). It is encouraging that the global average anomaly from the present is indistinguishable in the two studies, although one has to consider that the Shackleton et al estimate refers to the temperature of the whole ocean and not to its surface as here. What is the relationship between these two metrics at these timescales? This should be a fine opportunity to pick up the discussion on this in Shackleton et al, and see what else can be learned from the new global compilation, especially from the fact that, unlike from Hoffman et al., mean ocean temperatures don't seem here to much exceed global (or hemispheric?) SSTs. Also, it seems very important to understand how come the thermosteric implications for global sea levels are so much lower than obtained by both Shackleton et al and Hoffman et al? The latter use a relationship of 0.42-0.64 m °C−1 to infer a thermosteric contribution of 0.08-0.51 m. it is not clear how the authors obtained their thermosteric estimates.

---

## Referee Comment (RC1) · Anonymous Referee #1 · 13 Mar 2020

General comments

Turney et al. 2020 present an updated version of the Turney and Jones 2010 data compilation. As such, there is nothing too exciting about it but the inclusion of many new records, the effort to quantify ocean drift for all sites, and the resulting thermal expansion contribution to sea level are useful contributions and merit publication. There are similar data compilations (especially Hoffman et al. 2017) already to be found in the literature, with the main additional contribution of this work is the inclusion of more records and the quantification of ocean drift. Still, it is useful to see slightly different

approaches yielding generally similar results. The discussion of LIG sea surface temperatures is thus justifiably short, but the thermal expansion section could be fleshed out a bit more.

Specific comments

Turney et al. 2020 note that there are issues with previous approaches with regards to the reference period for all reported data, and they go on to express their anomalies as relative to modern instrumental observations. This seems like a reasonable thing to do, but it is difficult to estimate the effect of this change in referencing on the final data. It would be helpful and I would recommend to try to quantify the difference that arises from different referencing approaches, i.e. modern instrumental, preindustrial, or 20th century. This would allow closer comparison of this compilation to the works of Hoffman et al. 2017 and Capron et al. 2014.

As noted above, section 3.5 on thermal expansion could be substantially improved in my opinion. As already mentioned by Paolo Scussolini, the recent work of Shackleton et al. 2020 should be taken into account. Further, the methodology for computing the thermosteric contribution from sea surface data could be more detailed. It is stated that the top 700m of each grid cell is assumed to have changed according to the SST change. This seems like a fairly arbitrary depth that stems from the IPCC estimate for modern ocean warming (McKay et al. 2011). With the temperature anomaly estimates being very close to zero the volume used to calculate the thermisteric component is fairly irrelevant. Still, I would appreciate more justification or some sort of sensitivity of the final sea level numbers to the assumed ocean volume. Probably it's insignificant given the temperature dependence of the expansion coefficient, but would be interesting to see the thermosteric component if e.g. half the ocean volume warmed by the stated amount.

Finally, I have some issues with Table 1. The column headings need clarification, e.g. which latitude band does <45°S refer to? 23.5°S to 45°S, 0° to 45°S or something else? Same for <50°S. I'm not sure what the intention was with the order of the columns, but I would suggest going from the far north to the south and not switching back and forth between N and S. Furthermore, if Mean/uncorrected SST <45°S is 0.2 and Mean/uncorrected SST <50°S is 2.7, then the 45°S to 50°S latitude band must be very very warm (5+ degrees). Looking at Figure 4 or 5, this is not so. So something is off or I'm not understanding what is being shown in which case it should probably be described more clearly.

Technical corrections

Line 19: I recommend spelling out +6-11m as it is done in the main text to avoid confusion.

Line 58: Buizert et al. did not measure LIG $CO_2$ concentrations, I would suggest removing said citation.

Line 231: Should it say Figures 4 and 5?

Line 250: delete 'enable'.

Line 292: The NEEM community paper is a pure data paper, I don't see how that reference supports the preceding sentence.

Line 297: Buizert et al. also did not measure LIG sea level, hence that citation is inappropriate.

Line 410+: The bibliography also needs a bit of work. There are lots of links to nature.com supplementary information that should be removed and inconsistent usage of DOIs, some as full links, some as the number only.

---

## Referee Comment (RC2) · Anonymous Referee #2 · 21 Apr 2020

General comments:

The paper by Turney and others presents a new database of sea surface temperature (SST) data covering the Last Interglacial (129-116 kyr, LIG). The database presents 189 mean annual SST records, 99 DJF records, and 92 JJA records including faunal/ floral assemblages, Mg/Ca and Sr/Ca ratios, and alkenone Uk'37 proxies. The data are corrected for ocean drift using a model, a correction that has previously not been applied to SST data from the LIG. The authors do not present SST time series through the LIG or make an attempt to align SST records to a common chronological framework;

rather, they average the SST data across the period defined by the local minimum or "plateau" in benthic $\delta$18O, which they deem sufficiently representative of the LIG. During the LIG, mean global annual SST was +0.2 ± 0.1 °C warmer than modern. The compiled SST data are then used to quantify the thermosteric contribution to global mean sea level rise, which was +0.01 ± 0.1 m. The authors also examine the first 5 kyr during the LIG (129-124 kyr) to estimate the maximum SST and the thermosteric contribution to global mean sea level, found to be +0.9 ± 0.2 °C and +0.13 ± 0.1 m, respectively, relative to modern.

This dataset will potentially be valuable to other paleoclimate researchers and is well suited to be published in ESSD. However, I think the database would greatly benefit from more thorough presentation of the data in terms of their quality and their limitations (i.e. uncertainties therein and potential biases). For example, the data density (temporal resolution) for each record is not given or accounted for (n=? in each average SST value), the influence of outlier SST values on the LIG averages is not adequately addressed, and the spatial biases due to latitudinal/ longitudinal binning and/or lack of spatial resolution are not explored. The criteria for including records in the database need to be more rigorously and explicitly defined (were datasets rejected? how different is this compilation from the recent Hoffman 2017 compilation?).

Furthermore, the uncertainties acquired by applying the ocean drift correction are not addressed, nor are other models explored or tested to demonstrate model sensitivity.

Additionally, the authors attempted to avoid complications arising from chronological alignment of proxy records by averaging over the entire LIG period; however, there is zero discussion of how the $\delta$18O minimum was defined in each record, how well this minimum was expressed in their 203 different sites, or to what degree errors were inherited due to local variations in benthic $\delta$18O (even though the authors admit that such variations may temporally offset marine records by up to several millennia). In some cases, the SST records relied on proxies other than benthic $\delta$18O to define the LIG time period, but it is nowhere explained what alternative proxies were used, how

many records for which this was the case, or to what extent it might have influenced the results. The authors also do not address to what extent aligning the $\delta18O$ minima (because that is effectively what they are doing) warps the original age scales in the 203 records, except to show a very limited number of datasets (4) in Figure 2 – and there it is evident that the differences from the original age scales are substantial in some cases. Put another way, the authors need to address to what extent local variations in benthic $\delta18O$ might cause them to falsely identify the LIG time period and ultimately bias their LIG average temperature.

Finally, the manuscript would benefit from a comparison to other published LIG SST compilations (and estimates of thermosteric sea level rise) so that the reader either has some context for whether the new LIG reconstructions are reasonable, and/or why the new data are novel or represent an improvement on preexisting work. The authors also need to clarify what portion of the ocean volume their thermosteric sea level rise applies to (only surface 700 m?). It is confusing in the text as most of the authors' statements make it sound like whole ocean thermosteric sea level rise was calculated (I am still not 100 % certain).

If these comments can be sufficiently addressed, I see no reason not to publish this useful database. Please see specific comments below.

Specific comments (main text):

Line 109-110 – I cannot grasp how reliable this method was for selecting the LIG time period from the various proxy records based on what is presented in the manuscript. Were there any objective criteria for selecting $\delta18O$ minima? The authors must describe what they mean by "other complimentary proxy values," and state for how many records in the database this applies. The authors also must state what they mean by "such a $\delta18O$ plateau is not obvious." Were there objective criteria for electing to use alternative proxies rather than $\delta18O$? The authors seem to think spatial variations in $\delta18O$ are not an important source of error in their approach, though they admit below

that local variations can cause offsets of several millennia. Please provide more convincing arguments for this method and demonstrate to what extent these local $\delta$18O variations are important for your analyses.

Line 159-164 – The wording in this section is a bit too sleight of hand in my opinion. I disagree that the strategy is better than aligning records to a common temporal framework, or that it somehow circumvents the problem of generating time series data. While I agree that the authors do not interpret temporal trends (though they do distinguish the first 5 kyr from the rest of the LIG), by averaging over the selected periods with minimum $\delta$18O the authors are in essence still aligning records to a common chronology because their analysis assumes the periods were coeval. I also disagree that this strategy is better than the example of aligning North Pacific data with EDC $\delta$D (which they state could be off by 1-2 millennia) because Figure 2 shows even larger temporal offsets of up to $\sim$ 6 kyr (for example the end of the LIG in MD06-2986). The authors still need to present a convincing argument that aligning benthic $\delta$18O is robust against the spatio-temporal variability between sediment cores, and then please state some estimate of the uncertainty and inherited SST error.

Line 188-197 – Could you show some sensitivity analysis by running the model with different circulation? Just bracketing a plausible range would be enough to demonstrate the sensitivity. Also, I am very keen to see how the core top calibrations may change due to the ocean drift. I know the full analysis is beyond the scope of this paper, but perhaps selecting only a few core top measurements and examining how impacted they are by ocean drift would be useful for demonstrating the concept?

Line 203 – How is the uncertainty determined? If most proxies have uncertainties of 1-2 °C, it seems like the uncertainty on the mean should be larger than 0.1 °C.

Line 213 – So far I did not realize that you were just calculating the thermal expansion of the upper 700 m of the ocean. I highly recommend saying this in the text prior when stating your results (e.g. in the abstract and also in the introduction when discussing

previous sea level work). Otherwise, the reader may think you mean thermosteric sea level due to whole ocean thermal expansion (deep-water and surface).

Line 296-305 – Please specify here that the authors mean thermal expansion of the top 700 m of the ocean (which I think is what they mean, though it needs to be clarified more explicitly in the text). The authors should compare their result to other estimates of the thermosteric component of LIG sea level in addition to the McKay result (Hoffman et al., 2017;Shackleton et al., 2020).

Line 303-305 – This statement is too strong without explicitly stating that the deep ocean was not considered. Readers will misinterpret it to mean whole ocean thermosteric. Or, if the deep ocean was considered (I am still unclear about whether the authors did this or not), it must be justified why SST estimates alone were used to estimate whole ocean thermosteric sea level rise and why the estimates were so low compared to other work (e.g. Shackleton 2020).

Figure 2 – Showing the alignment of only four marine cores is much too limited to give readers any sense for how much the 203 chronologies were distorted when the authors picked $\delta$18O minima to delineate the LIG time period, over which they averaged the SST results. Figure 2 demonstrates that for none of the four cores shown did the LIG actually occur during the period 129-116 kyr (on their respective age models), and in core MD06-2986 the LIG notably occurred during a span of only about 5 kyr. Can you say with confidence (or even better, demonstrate for readers) that the cores in Figure 2 represent the full range of chronological differences in the $\delta$18O minima between all of the records? Additionally, please improve the figure resolution so that the text and traces are not blurry.

Figure 3 – This is confusing. It looks like only the modern data were run through the drift correction. I thought the correction was applied to each LIG average.

Figure 4 – I recommend plotting a third panel showing the residual between the original SST and the drift-corrected SST.
Table 1 – It strikes me as odd that the DJF and JJA global SST values are both negative, whereas the mean global SST value is positive. What delineated a DJF and JJA record from the other 189 records? How much overlap is there between the 92 + 99 seasonal records and the 189 annual records?

Table 2 – Similar comment as above.

Specific comments (regarding the Excel file):

Sheet 1 – The spatial delineations are confusing. Why do you average > 45° and then also > 50° with only 5° difference? Please justify.

Column H - By "Jan-Dec" do you mean annual? Just say "annual" so as not to be confused with "DJF."

Technical corrections:

Line 42 – "The timing and impacts. . . remain. . ." instead of "remains."

Line 47 – Better references exist for "multi-millennial duration shifts in the Earth system took place in the past." The ones used here appear to mostly be about Anthropocene/ future tipping points.

Line 51 – Can you provide a reference for 129,000-116,000 years ago, if it is elsewhere defined? Otherwise state it is the authors' definition.

Line 56 – Global Mean Sea Level should not be capitalized.

Line 57 – There are better references for the observation of abrupt shifts in regional hydroclimate during the last interglacial than Thomas et al. 2015. Why not just cite cave record papers (Wang et al., 2008;Cheng et al., 2016), for example?

Line 58 – Buizert 2014 is not about CO2. Kohler 2017 is partly, but why not cite the original data? (Petit et al., 1999;Barnola et al., 1987) or (Bereiter et al., 2015) for the most recent compilation of CO2 ice core data.

[Figure]

Line 61 – Provide references for "considerable debate" about the contribution of sources to sea level rise.

Line 74 – Cite also (Hoffman et al., 2017).

Line 80 – Sea-Surface Temperature should not be capitalized.

Line 83 – Can you move the Mercer 1978 reference to somewhere in the middle of the sentence? At the end of the sentence it looks like it is a reference for the Paris Climate Agreement.

Lilne 117 – Does "maximum" refer to the average of the first 5kyr? I recommend changing the wording because "maximum" can be interpreted here that your means are upper limits.

Line 121-123 – I don't think Figure 3 should be referenced here, as it doesn't really relate to what is said in the sentence.

Line 125-129 – Again the use of the word "maximum" could be misunderstood to mean you only used the highest values in the datasets, especially on line 126.

References:

Barnola, J. M., Raynaud, D., Korotkevich, Y. S., and Lorius, C.: Vostok ice core provides 160,000-year record of atmospheric CO2, Nature, 329, 408-414, 10.1038/329408a0, 1987.

Bereiter, B., Eggleston, S., Schmitt, J., Nehrbass-Ahles, C., Stocker, T. F., Fischer, H., Kipfstuhl, S., and Chappellaz, J.: Revision of the EPICA Dome C CO2 record from 800 to 600kyr before present, Geophysical Research Letters, 42, 542-549, 10.1002/2014gl061957, 2015.

Cheng, H., Edwards, R. L., Sinha, A., Spotl, C., Yi, L., Chen, S. T., Kelly, M., Kathayat, G., Wang, X. F., Li, X. L., Kong, X. G., Wang, Y. J., Ning, Y. F., and Zhang, H. W.: The Asian monsoon over the past 640,000 years and ice age terminations, Nature, 534,

640-+, 10.1038/nature18591, 2016.

Hoffman, J. S., Clark, P. U., Parnell, A. C., and He, F.: Regional and global sea-surface temperatures during the last interglaciation, Science, 355, 276-279, 10.1126/science.aai8464, 2017.

Petit, J. R., Jouzel, J., Raynaud, D., Barkov, N. I., Barnola, J. M., Basile, I., Bender, M., Chappellaz, J., Davis, M., Delaygue, G., Delmotte, M., Kotlyakov, V. M., Legrand, M., Lipenkov, V. Y., Lorius, C., Pepin, L., Ritz, C., Saltzman, E., and Stievenard, M.: Climate and atmospheric history of the past 420,000 years from the Vostok ice core, Antarctica, Nature, 399, 429-436, 10.1038/20859, 1999.

Shackleton, S., Baggenstos, D., Menking, J. A., Dyonisius, M. N., Bereiter, B., Bauska, T. K., Rhodes, R. H., Brook, E. J., Petrenko, V. V., McConnell, J. R., Kellerhals, T., Haberli, M., Schmitt, J., Fischer, H., and Severinghaus, J. P.: Global ocean heat content in the Last Interglacial, Nature Geoscience, 13, 7, 10.1038/s41561-019-0498-0, 2020.

Wang, Y. J., Cheng, H., Edwards, R. L., Kong, X. G., Shao, X. H., Chen, S. T., Wu, J. Y., Jiang, X. Y., Wang, X. F., and An, Z. S.: Millennial- and orbital-scale changes in the East Asian monsoon over the past 224,000 years, Nature, 451, 1090-1093, 10.1038/nature06692, 2008.

---

## Referee Comment (RC3) · Jeremy Hoffman (Referee) · 7 May 2020

Turney et al. have compiled the most comprehensive data base of sea-surface temperatures spanning the last interglaciation (LIG) to date. Their results support the conclusions of several recent studies in important ways, even given their (novel) attention to potentially confounding effects present within SST reconstructions from planktonic sources (their "ocean drift") that were largely unaddressed in previous LIG work.

Understandably there has been considerable attention to the LIG as it can serve to

assess the sensitivity of important Earth systems (such as the cryosphere, which was considerably smaller than at present due to higher insolation and warmer global temperatures) to natural climate fluctuation in recent Earth history, potentially illuminating mechanisms currently unaccounted for or underestimated in present-day climate models.

Having a "living repository" of LIG datasets from the marine realm will do well to improve future (and ongoing) LIG model-data comparisons, as is highlighted by the authors. The accompanying article is appropriate to support the publication of this dataset. The dataset is highly useful, unique in its comprehensive nature, and functionally complete. This dataset is of extremely high quality.

However, Turney et al. add only marginally to the existing story about total LIG warming amplitude relative to recent climatology (their uncertainties on a global anomaly overlap with basically all previous work!) and, by their chosen study design, can't add anything to the discussions ongoing about rates, extents, and locations of warming or sea-level change at particular times within the LIG. These stories have recently been borne a bit more out of work in modeling (Clark et al., 2020, Nature - referenced below) and a new ice-core based SST reconstruction (Shackleton et al., 2020, Nature Geoscience).

I am curious how the authors can work on an update to the manuscript that incorporates more discussion of the understanding of intra-LIG variability in sea level, temperature, and other variables, and as such, work to clearly justify just why the multi-millennial, LIG-long averages that they have generated help us to better understand those variables or model outputs. Are there modeling studies planned (lig127k PMIP?) that they can point to that would be targets for comparison with their new reconstruction? If the main SST magnitude conclusions aren't different from previous work, and the work can't resolve anything particularly new within the LIG time period, maybe the effort of the paper should simply focus on updating the maximum possible thermosteric component of LIG sea level and make that the centerpiece of the analysis?

Specific comments -

Lines 188-197 – Are the ocean drift correction calculations estimated using the HadISST data used to calculate the anomalies from climatology as well? How are these "life trajectory" SST averages (which presumably have some sort of standard deviation or variance across space/time) then incorporated into the SST reconstruction uncertainty? Addressing this additional source of uncertainty in the SST estimates may further complicate the story that arises from the drift-corrected SSTs, but perhaps maybe only subtly. This might be worthwhile discussing or exploring in a couple of particular locations, especially those where the signals due to drift correction are large. I would suspect that as these areas have large SST gradients themselves that estimating an "average" SST across their lifetime/drift might generate some additional uncertainty in the estimated anomaly.

Lines 63-68 – please add Clark, P.U., He, F., Golledge, N.R. et al. Oceanic forcing of penultimate deglacial and last interglacial sea-level rise. Nature 577, 660–664 (2020). https://doi.org/10.1038/s41586-020-1931-7 to references about ice sheet modeling during this time period, as well as amounts from particular reservoirs/sources of sea-level rise. Given these recent estimates of intra-LIG sea-level change (citations within), what does this "maximum" LIG thermosteric component tell us?

Discussion of the LIG-long averages and addressing the small specific considerations would, in my mind, improve the clarity of this largely incremental - however important! - addition to the body of LIG SST knowledge. I thank the authors for the opportunity to comment and look forward to reading an updated draft of the manuscript.

———————————————————

---

## Author Comment (AC1) · 14 Sep 2020

**Response to Reviewers Comments (essd-2019-249)**

**REVIEWER #SC1 (PAOLO SCUSSOLINI)**

This is a welcome work that tackles a key question that is presently still insufficiently resolved: understanding global and regional temperatures during a key instance of past warm climate. It is ideal that independent groups of researchers address the same problem with different approaches and producing comparable results, something that also addresses the hotly discussed issue of reproducibility in the sciences at large. This study parallels a number of previous efforts, and most closely the recent work of Hoffman et al (2017). The main differences with that study are, in subjective order of importance: ocean drift correction is applied; SSTs are integrated across the whole LIG; a larger sample of SST proxy records; much larger sampling of seasonal SSTs.

We thank the reviewer for their kind words and recognition of the value of this study. As Reviewer #SC1 highlights, this study provides a contribution to an important topic: the sensitivity of the Earth system to relatively high temperatures during past interglacials. In contrast to other studies, this study makes several contributions including a study into the potential role of ocean drift in reconstructing Last Interglacial temperatures, the development of a robust reconstruction of mean temperatures, the largest yet published network of quantified sea surface temperatures, and an analysis of published seasonal SSTs.

The accounting of the oceanographic footprint of the proxy records seems to me the clearest novelty introduced in this work. This is very timely, and the importance of the drift is clear as seen in the biases in Fig. 1, although I expected this to also impact the global SST estimate. The authors provide some sensitivity test on the choice of the lifespan parameter of the virtual particles, but I find this aspect somewhat incomplete, as it focused only on parameters appropriate for foraminifera. In a sensitivity test, only lifespans longer than the 30-day value adopted in the database are tested, while shorter lifespans seem plausible for coccolith-based reconstructions, which make up much of the database; the sinking speed of 200 m/day and the 30 m depth for the lifecycle may not be adequate to simulate the situation with coccoliths and other organisms smaller than foraminifera, and with phytoplankton that is confined to the photic zone. I am not expert in these organisms, but it should have been relatively easy to apply dif- ferent parameters to the main type of organisms relevant to the database (that is, if the literature suggests that these are substantially different from those used), and at least test the effect of taking unique values for the whole database when a differentiation could have been possible. Also, while this probably exceeds the scopes of this study, would it be possible to mention why a simulation of OFES with LIG boundary conditions is not contemplated, e.g., initiated with data from the coarser grid of an ocean model from a PMIP4 GCM? Maybe an idea for future work.

We thank the reviewer for their comments regarding the lifespan of different organisms. For sure, there will almost certainly be an effect from different lifespans (and sinking rates) but that is a considerable expansion in the scope of the study from this initial investigation. Our intention in this work was to explore whether the amount of drift using contemporary ocean dynamics was sufficient to cause a substantial difference in regional and global temperature estimates. In this study we find that some sectors record relatively large anomalously warm

signals, up to 3.5°C, for example in the tropical East Pacific, the North Atlantic and South China Sea. Future work will investigate the impact of drift on different taxa for temperature reconstruction. This work would ideally also use an eddy-resolved Last Interglacial model simulation to quantify the lateral advection of sinking particles. Unfortunately, recent work by EvS and colleagues (Nooteboom *et al.*, 2020, *PlosOne*), has demonstrated that palaeoclimate modelling simulations generally have insufficient spatial resolution to capture mesoscale features that are critical for modelling particle drift. We hope future modelling outputs will enable this work to be undertaken. As a result, in the revised manuscript, we have acknowledged that the drift is estimated by contemporary ocean circulation which we consider to be a reasonable first-order approximation of Last Interglacial conditions. Reference: Nooteboom, P.D., Delandmeter, P., van Sebille, E., Bijl, P.K., Dijkstra, H.A., von der Heydt, A.S., 2020. Resolution dependency of sinking Lagrangian particles in ocean general circulation models. *PLoS ONE* 15, e0238650.

The integration of SSTs across the whole period has both advantages and pitfalls: on the one hand it makes results independent from the delicate set of choices that necessarily come with assessing age models and aligning them within and across basins on a coherent chronology; on the other hand it dismisses the millennial scale variability that is critical to understand notable climatic variability within the LIG. The authors recognize this, but I suggest that a more convincing explanation could be provided of the choice of working from the hypothesis (as in Turney and Jones 2010) of global synchronicity of peak SSTs: why is it superior to other solutions that make some use of the each record's explicit age models, what are the implications of the assumption for the results?

The reviewer is absolutely correct that it is a delicate balance resolving the numerous chronological uncertainties of individual sedimentary records with robust millennial-scale reconstructions possible in some records. Most studies rely on some form of alignment that link sequences to one or more reference records with robust chronological frameworks. As Hoffman *et al.* (2017) demonstrated, the age uncertainties remain considerable for the Last Interglacial (up to several millennia during the LIG e.g. their Fig S7). Here, the authors aligned marine records to speleothem-dated, ice core reconstructions, assuming synchronous climate changes in the records. This approach is not without its problems, however.  More than half of reported Pacific marine cores (from the Northern Hemisphere) were correlated to the Antarctic EPICA Dome C dD record (page 3 of our manuscript) even though this study highlighted that the south leads the warming of the north by 1-2 millennia. The development of accurate and precise age estimates for the LIG is urgently needed to resolve the timing of global climate change but will require a considerable future international effort. We have provided a more detailed explanation of our approach on pages 3 and 4 of the manuscript. We stress we do not wish to underplay the importance of resolving millennial-scale variability in the climate evolution of the LIG but this is not the focus of this study. Here we are using the mean temperature estimates to constrain the role of thermal expansion in global sea level rise across the LIG, and also provide boundary conditions for future modelling studies investigating the impact of warming on polar ice sheets. Whilst we may sacrifice temporal control, our study does help minimise the uncertainty on zonal and global temperature averages.

Last, it is important that the results are discussed in the light of the new results on mean LIG ocean temperature based on Antarctic noble gas, in the paper by Shackleton et al. just out in January (2020; doi: 10.1038/s41561-019-0498-0). It is encouraging that the global average anomaly from the present is indistinguishable in the two studies, although one has to consider that the Shackleton et al estimate refers to the temperature of the whole ocean and not to its surface as here. What is the relationship between these two metrics at these timescales? This should be a fine opportunity to pick up the discussion on this in Shackleton et al, and see what else can be learned from the new global compilation, especially from the fact that, unlike from Hoffman et al., mean ocean temperatures don't seem here to much exceed global (or hemispheric?) SSTs. Also, it seems very important to understand how come the thermosteric implications for global sea levels are so much lower than obtained by both Shackleton et al and Hoffman et al? The latter use a relationship of 0.42-0.64 m $\circ$C$^{-1}$ to infer a thermosteric contribution of 0.08-0.51 m. it is not clear how the authors obtained their thermosteric estimates.

We thank the reviewer for highlighting the importance of the Shackleton *et al*. paper. This was published after our submission to the journal and is now part of the discussion in our revised manuscript. As the reviewer states, the new work by Shackleton and colleagues uses noble gas measurements from Antarctic ice cores (Taylor Glacier and EPICA Dome C). The isotopic ratios in atmospheric trace gas (nitrogen, xenon and krypton) are sensitive to the mean ocean temperature via their solubility in seawater. These results suggest an early LIG peak in ocean heat content contributed 0.7±0.3 m, subsequently declining to no appreciable contribution after 127 kyr. In contrast, Hoffman et al., reported a range of 0.08 to 0.51 m for peak (early) LIG warmth centred on 125 kyr (although this is after 127 kyr reported by Shackleton et al. this is almost certainly the same event but represents the age uncertainties in the marine records). Here we have not attempted to resolve the relative timing of peak warmth but have determined the maximum temperature within the first 5 kyr of the Last Interglacial to provide an upper estimate of the contribution from thermal expansion. In the revised manuscript we have provided more detail on how we calculated the thermosteric sea level rise. In our previous submitted version of the manuscript, we followed the procedure reported by McKay et al (2011). To provide a maximum estimate of thermosteric sea level rise, we assumed our average SST warming was representative of the uppermost 700 m in the water column. Using the Thermodynamic Equation of Seawater 2010 (TEOS-10) we calculated the change in the specific volume of the upper 700 m of the ocean while holding the salinity constant, and neglecting changes in ocean area. We determined the change in the specific volume of the top 700 m of each a 10° latitude × 10° grid cell while holding the salinity constant. As the reviewer hints, it is possible that sustained warming ocean occurred below 700 m. We have therefore repeated the above analysis down to an average ocean depth of 2000 m (approximately the upper half of the ocean) and 3500 m (the whole ocean). The results for the early LIG are as follows:

700 m depth of warming: GMSL of 0.12 ± 0.10 m (uncorrected) and 0.13 ± 0.10 m (drift corrected).

2000 m depth of warming: GMSL of 0.36 ± 0.10 m (uncorrected) and 0.39 ± 0.10 m (drift corrected).

3500 m depth of warming: GMSL of 0.67 ± 0.10 m (uncorrected) and 0.72 ± 0.10 m (drift corrected).

Thus, our reconstructed SSTs suggest a mean thermosteric sea level rise of 0.08 ± 0.1 m and a maximum of 0.39 ± 0.1 m respectively (assuming warming penetrated to 2000 m depth). These estimates provide upper limits on thermosteric sea level rise. Our results are consistent with the absolute amount and timing of the contribution reported by Shackleton et al. (2020) and Hoffman et al. (2017). We have included these new results in our revised manuscript, highlighting the results from 2000 m water depth as the more likely scenario. The revised figures 5 (mean annual across the full Last Interglacial) and 6 (maximum temperatures during the early Last Interglacial) are provided below.

[Figure]

**Figure 5:** Global and zonal mean annual sea-surface temperature (SST) anomalies and thermosteric sea level change across the full Last Interglacial. Temperature anomalies reported as uncorrected (panels a and c respectively) and after applying 30-day (panels b and d respectively) temperature offsets arising from ocean current drift. Uncertainty for zonal average reconstructions given at 1sd. Here ocean warming is assumed to have penetrated to 2000 m depth, on average. Temperature estimates relative to the modern period (CE 1981-2010).

[Figure]

**Figure 6:** Global and zonal mean annual sea-surface temperature (SST) anomalies and thermosteric sea level change during the early Last Interglacial. Temperature anomalies reported as uncorrected (panels a and c respectively) and after applying 30-day (panels b and d respectively) temperature offsets arising from ocean current drift. Uncertainty for zonal average reconstructions given at 1sd. Here ocean warming is assumed to have penetrated to 2000 m depth, on average. Temperature estimates relative to the modern period (CE 1981-2010).

Our analysis allows us to identify the geographic contributions of thermal expansion to sea level. These figures show the zonal contributions of the maximum thermostatic sea level contribution were greatest at high latitudes, and were negligible (or possibly even negative) in the tropics, an observation not previously made in the literature. We have now made an explicit statement that there was an early peak contribution from thermal expansion during the early interglacial (something that was missing from the previous submission), further highlighting the important contribution polar ice melt must have made to account for the known substantial sea level height throughout the LIG.

---

## Author Comment (AC2) · 15 Sep 2020

**Response to Reviewers Comments (essd-2019-249)**

**REVIEWER #1**

Turney et al. 2020 present an updated version of the Turney and Jones 2010 data compilation. As such, there is nothing too exciting about it but the inclusion of many new records, the effort to quantify ocean drift for all sites, and the resulting thermal expansion contribution to sea level are useful contributions and merit publication. There are similar data compilations (especially Hoffman et al. 2017) already to be found in the literature, with the main additional contribution of this work is the inclusion of more records and the quantification of ocean drift. Still, it is useful to see slightly different approaches yielding generally similar results. The discussion of LIG sea surface temperatures is thus justifiably short, but the thermal expansion section could be fleshed out a bit more.

As Reviewer #SC1 highlights, there are several major innovations in this study. In contrast to other studies, this study makes several contributions including a study into the potential role of ocean drift in reconstructing Last Interglacial temperatures, the development of a robust reconstruction of mean temperatures, the largest yet published network of quantified sea surface temperatures, and an analysis of published seasonal SSTs. As Reviewer #1 acknowledges, it is valuable that different approaches for reconstructing LIG temperatures show broadly consistent results, providing increased confidence in our understanding of the sensitivity of the Earth system to high temperatures.

Specific comments

Turney et al. 2020 note that there are issues with previous approaches with regards to the reference period for all reported data, and they go on to express their anomalies as relative to modern instrumental observations. This seems like a reasonable thing to do, but it is difficult to estimate the effect of this change in referencing on the final data. It would be helpful and I would recommend to try to quantify the difference that arises from different referencing approaches, i.e. modern instrumental, preindustrial, or 20th century. This would allow closer comparison of this compilation to the works of Hoffman et al. 2017 and Capron et al. 2014.

The use of different time periods to represent 'present day' has somewhat confused the literature. Whilst we appreciate the sentiment of the reviewer, there are major problems with using earlier periods (e.g. pre-industrial) to express relative temperature differences given the long known and continuing paucity of observations further back in time, particularly in remote locations e.g. Brohan et al., 2006. Such a study would need to fully quantify the uncertainties in the limited network of 'observations' prior to the satellite era, only increasing the uncertainties further, and would be a separate study in itself. As a result we are concerned this may further confuse the literature and are hesitant to undertake comparisons as suggested by the reviewer. We hope the Editor approves. Reference: Brohan, P., Kennedy, J.J., Harris, I., Tett, S.F.B., Jones, P.D., 2006. Uncertainty estimates in regional and global observed temperature changes: A new data set from 1850. Journal of Geophysical Research 111, D12106.

As noted above, section 3.5 on thermal expansion could be substantially improved in my opinion. As already mentioned by Paolo Scussolini, the recent work of Shackleton et al. 2020 should be taken into account. Further, the methodology for computing the thermosteric contribution from sea surface data could be more detailed. It is stated that the top 700m of each grid cell is assumed to have changed according to the SST change. This seems like a fairly arbitrary depth that stems from the IPCC estimate for modern ocean warming (McKay et al. 2011). With the temperature anomaly estimates being very close to zero the volume used to calculate the thermisteric component is fairly irrelevant. Still, I would appreciate more justification or some sort of sensitivity of the final sea level numbers to the assumed ocean volume. Probably it's insignificant given the temperature dependence of the expansion coefficient, but would be interesting to see the thermosteric component if e.g. half the ocean volume warmed by the stated amount.

We thank the reviewer for their suggestion. We have expanded the discussion on the thermosteric sea level rise as Reviewer #SC1 suggested. And following on from the recommendation of this review we have included the analysis of the greater ocean depths (2000 m and 3500 m). We derived the following results:

2000 m depth of warming: GMSL of 0.36 ± 0.10 m (uncorrected) and 0.39 ± 0.10 m (drift corrected).

3500 m depth of warming: GMSL of 0.67 ± 0.10 m (uncorrected) and 0.72 ± 0.10 m (drift corrected).

We thank the reviewer for the suggestion. We have now also expanded the discussion to include Shackleton et al. (2020) paper which was published after our submission.

Finally, I have some issues with Table 1. The column headings need clarification, e.g. which latitude band does <45∘S refer to? 23.5∘S to 45∘S, 0∘ to 45∘S or something else? Same for <50∘S. I'm not sure what the intention was with the order of the columns, but I would suggest going from the far north to the south and not switching back and forth between N and S. Furthermore, if Mean/uncorrected SST <45∘S is 0.2 and Mean/uncorrected SST <50∘S is 2.7, then the 45∘S to 50∘S latitude band must be very very warm (5+ degrees). Looking at Figure 4 or 5, this is not so. So something is off or I'm not understanding what is being shown in which case it should probably be described more clearly.

We must apologise. Looking at the table again, we realised it was confusing. The four columns in question refer to polewards of either 45˚ or 50˚ in both hemispheres. We have now made this explicit and reordered the columns as the reviewer has recommended.

Technical corrections
Line 19: I recommend spelling out +6-11m as it is done in the main text to avoid confusion.
Done.

Line 58: Buizert et al. did not measure LIG CO2 concentrations, I would suggest removing said citation.
This study did report CO2 concentrations from Taylor Glacier but we are happy to remove the citation.

Line 231: Should it say Figures 4 and 5? Line 250: delete 'enable'.
Done.

Line 292: The NEEM community paper is a pure data paper, I don't see how that reference supports the preceding sentence.
Done.

Line 297: Buizert et al. also did not measure LIG sea level, hence that citation is inappropriate.
We apologise. This has been removed.

Line 410+: The bibliography also needs a bit of work. There are lots of links to nature.com supplementary information that should be removed and inconsistent usage of DOIs, some as full links, some as the number only.
We have edited the references to tidy them up. Sorry about this.

---

## Author Comment (AC3) · 15 Sep 2020

**Response to Reviewers Comments (essd-2019-249)**

**REVIEWER #2**
This dataset will potentially be valuable to other paleoclimate researchers and is well suited to be published in ESSD. However, I think the database would greatly benefit from more thorough presentation of the data in terms of their quality and their limitations (i.e. uncertainties therein and potential biases). For example, the data density (temporal resolution) for each record is not given or accounted for (n=? in each average SST value), the influence of outlier SST values on the LIG averages is not adequately addressed, and the spatial biases due to latitudinal/ longitudinal binning and/or lack of spatial resolution are not explored. The criteria for including records in the database need to be more rigorously and explicitly defined (were datasets rejected? how different is this compilation from the recent Hoffman 2017 compilation?).

Because we are not investigating centennial and millennial-scale variability, we were able to expand the number of records to that reported by Hoffman et al. The key criteria was that there was a minimum of three SST estimates across the LIG. In contrast, Hoffman et al. was focussed on time series data that required: 'The sample resolution ranges from centennial to <4000 years on their published age models, with a median resolution of 1100 years.' We are therefore able to report almost double the number of records to that presented by Hoffman et al. (189 vs 104 mean annual SSTs). Inevitably there are differences in the number of analyses undertaken through the different records which is dependent on the accumulation rate. In addition to the large database of temperature reconstructions, in response to the reviewer's suggestion, we now include the temporal density of the SST observations. For the error calculated for the regional and global SST anomalies, we incorporate the errors from the SST proxies (reported in the database), and the error associated with estimating regional and global SST from limited spatial coverage. To achieve this we propagated the SST errors for each measurement through each of the averaging steps (i.e. temporal to grid cell to zonal to area-weighted global) in our ocean-area-weighted average, as described by McKay et al. (2011). We used quoted error estimates for each study where reported. If not available, we applied proxy-specific error estimates. Although the impact of the spatial coverage was not explored in this study, it has been previously estimated in McKay et al., 2011. In that study, the error associated with the limited spatial range of the oceanographic proxies was estimated by calculating 1000 random 1-year global SST anomalies over the twentieth century, and comparing that to averages derived using only the paleoceanographic network available to that study. With that approach, they found no systematic biases associated with spatial network, and a 1 sigma uncertainty estimate of <0.1 degree. In this study, we've expanded the spatial network, and so it's reasonable to to consider ±0.1 degrees Celsius a reasonable, high-end estimate, making the contribution of spatial uncertainty modest in comparison to the other uncertainties in the study.

Furthermore, the uncertainties acquired by applying the ocean drift correction are not addressed, nor are other models explored or tested to demonstrate model sensitivity.
The full method of the ocean drift is provided by van Sebille et al. (2015). This approach tracks virtual particles in an eddy-resolving ocean model, the Japanese Ocean model For

the Earth Simulator or OFES. In future work we would like to explore other models. In our previous work, however, we utilised the INALT01 model and found the ±1 s.d. of the INALT01, OFES and proxy distributions overlap. See figure below using two examples.   We therefore consider the OFES to provide a robust estimate of possible drift in this early study.

[Figure]

**Figure from van Sebille et al. (2015):** Distributions of temperature at two cores in the Agulhas region. The observed proxy temperatures (grey bars) at (a) the Agulhas Current core and (b) the Agulhas leakage core are compared with the temperature distributions for the virtual foraminifera experiments in the INALT01 model (red) and the OFES model (blue).

Reference: van Sebille, E., Scussolini, P., Durgadoo, J.V., Peeters, F.J.C., Biastoch, A., Weijer, W., Turney, C., Paris, C.B., Zahn, R., 2015. Ocean currents generate large footprints in marine palaeoclimate proxies. Nature Communications 6, 6521.

Additionally, the authors attempted to avoid complications arising from chronological alignment of proxy records by averaging over the entire LIG period; however, there is zero discussion of how the δ18O minimum was defined in each record, how well this minimum

was expressed in their 203 different sites, or to what degree errors were inherited due to local variations in benthic δ18O (even though the authors admit that such variations may temporally offset marine records by up to several millennia). In some cases, the SST records relied on proxies other than benthic δ18O to define the LIG time period, but it is nowhere explained what alternative proxies were used, how many records for which this was the case, or to what extent it might have influenced the results. The authors also do not address to what extent aligning the δ18O minima (because that is effectively what they are doing) warps the original age scales in the 203 records, except to show a very limited number of datasets (4) in Figure 2 – and there it is evident that the differences from the original age scales are substantial in some cases. Put another way, the authors need to address to what extent local variations in benthic δ18O might cause them to falsely identify the LIG time period and ultimately bias their LIG average temperature.

As the reviewer correctly identifies no one method provides an absolute age model for the last Interglacial. Even the use of d18O to define the LIG has an age uncertainty of 1-2 millennia. In some records where d18O was unavailable, other proxies used by the original authors have been used to identify the placement of the LIG; for instance,  the CaCO3 content of the sediments as a measure of glacial-interglacial variability. However, it is important to note that we are not aiming to resolve centennial and millennial-scale variability through the interglacial and while we acknowledge that some individual SST estimates may not fall within the LIG or have been excluded (due to the uncertainties in the d18O for defining the interglacial) we consider the averaging of values across the full interglacial provides a robust value for each record and ultimately the regional and global reconstructions.

Finally, the manuscript would benefit from a comparison to other published LIG SST compilations (and estimates of thermosteric sea level rise) so that the reader either has some context for whether the new LIG reconstructions are reasonable, and/or why the new data are novel or represent an improvement on preexisting work. The authors also need to clarify what portion of the ocean volume their thermosteric sea level rise applies to (only surface 700 m?). It is confusing in the text as most of the authors' statements make it sound like whole ocean thermosteric sea level rise was calculated (I am still not 100 % certain).

We have now expanded the discussion of how we calculated the thermosteric sea level rise. As the reviewer correctly surmised we had originally determined this for the uppermost 700 m of the ocean. But we have now expanded the analysis to include the uppermost 2000 metres (approximately half the world's ocean) and 3500 metres. The 2000 metre depth warming provides comparable results to those reported by Shackleton et al (2020) and Hoffman et al  (2017) which we have now discussed in the text.

If these comments can be sufficiently addressed, I see no reason not to publish this useful database.
We thank the reviewer for their support.

Specific comments (main text):
Line 109-110 – I cannot grasp how reliable this method was for selecting the LIG time period from the various proxy records based on what is presented in the manuscript. Were there any objective criteria for selecting δ18O minima? The authors must describe what they mean by "other complimentary proxy values," and state for how many records in the database this

applies. The authors also must state what they mean by "such a δ18O plateau is not obvious." Were there objective criteria for electing to use alternative proxies rather than δ18O? The authors seem to think spatial variations in δ18O are not an important source of error in their approach, though they admit below that local variations can cause offsets of several millennia. Please provide more convincing arguments for this method and demonstrate to what extent these local δ18O variations are important for your analyses.

We have addressed this issue in the main manuscript by explicitly recognising the uncertainties in the recognition of the d18O minima (and other proxies such as CaCO3) in each record, stating the uncertainty in this method and emphasizing the averaging of values across the full interglacial provides a robust value for each record and ultimately the regional and global reconstructions (see above).

Line 159-164 – The wording in this section is a bit too sleight of hand in my opinion. I disagree that the strategy is better than aligning records to a common temporal framework, or that it somehow circumvents the problem of generating time series data. While I agree that the authors do not interpret temporal trends (though they do distinguish the first 5 kyr from the rest of the LIG), by averaging over the selected periods with minimum δ18O the authors are in essence still aligning records to a common chronology because their analysis assumes the periods were coeval. I also disagree that this strategy is better than the example of aligning North Pacific data with EDC δD (which they state could be off by 1-2 millennia) because Figure 2 shows even larger temporal offsets of up to ~ 6 kyr (for example the end of the LIG in MD06-2986). The authors still need to present a convincing argument that aligning benthic δ18O is robust against the spatio-temporal variability between sediment cores, and then please state some estimate of the uncertainty and inherited SST error.

The age models reported in Figure 2 are from the original studies. We have not attempted to generate new age models. We are simply recognising the LIG in each record and then averaging the SST estimates over what we consider to be a common time period. The statement about the alignment of North Pacific data with the Antarctic EDC δD was to emphasise the challenges of identifying asynchronous changes between the hemispheres. Here we take a different approach to derive a first-order estimate of the temperature through the Last Interglacial, bypassing such issues.

Line 188-197 – Could you show some sensitivity analysis by running the model with different circulation? Just bracketing a plausible range would be enough to demonstrate the sensitivity. Also, I am very keen to see how the core top calibrations may change due to the ocean drift. I know the full analysis is beyond the scope of this paper, but perhaps selecting only a few core top measurements and examining how impacted they are by ocean drift would be useful for demonstrating the concept?

Unfortunately, recent work by EvS and colleagues (Nooteboom *et al*., 2020, *PlosOne*), has demonstrated that palaeoclimate modelling simulations have insufficient spatial resolution to capture mesoscale features that are critical for modelling particle drift. We hope future

modelling outputs will enable this work to be undertaken. As a result, in the revised manuscript, we have acknowledged that the drift is estimated by contemporary ocean circulation which we consider to be a reasonable first-order approximation of the Last Interglacial. In future work we would like to undertake a detailed study of the impact of drift on the calibration but such a study would be beyond the scope of this database. We hope by highlighting the potentially substantial impact of drift (particularly in some key locations) this may be a focus for future research for others in the community as well. Reference: Nooteboom, P.D., Delandmeter, P., van Sebille, E., Bijl, P.K., Dijkstra, H.A., von der Heydt, A.S., 2020. Resolution dependency of sinking Lagrangian particles in ocean general circulation models. *PLoS ONE* 15, e0238650.

Line 203 – How is the uncertainty determined? If most proxies have uncertainties of 1-2 ◦C, it seems like the uncertainty on the mean should be larger than 0.1 ◦C.
We have described this more fully in the revised manuscript.

Line 213 – So far I did not realize that you were just calculating the thermal expansion of the upper 700 m of the ocean. I highly recommend saying this in the text prior when stating your results (e.g. in the abstract and also in the introduction when discussing previous sea level work). Otherwise, the reader may think you mean thermosteric sea level due to whole ocean thermal expansion (deep-water and surface).
Done. We apologise for the confusion.

Line 296-305 – Please specify here that the authors mean thermal expansion of the top 700 m of the ocean (which I think is what they mean, though it needs to be clarified more explicitly in the text). The authors should compare their result to other estimates of the thermosteric component of LIG sea level in addition to the McKay result (Hoffman et al., 2017;Shackleton et al., 2020).

Done.

Line 303-305 – This statement is too strong without explicitly stating that the deep ocean was not considered. Readers will misinterpret it to mean whole ocean thermosteric. Or, if the deep ocean was considered (I am still unclear about whether the authors did this or not), it must be justified why SST estimates alone were used to estimate whole ocean thermosteric sea level rise and why the estimates were so low compared to other work (e.g. Shackleton 2020).

Done. We apologise for the confusion.

Figure 2 – Showing the alignment of only four marine cores is much too limited to give readers any sense for how much the 203 chronologies were distorted when the authors picked δ18O minima to delineate the LIG time period, over which they averaged the SST results. Figure 2 demonstrates that for none of the four cores shown did the LIG actually

occur during the period 129-116 kyr (on their respective age models), and in core MD06-2986 the LIG notably occurred during a span of only about 5 kyr. Can you say with confidence (or even better, demonstrate for readers) that the cores in Figure 2 represent the full range of chronological differences in the δ18O minima between all of the records? Additionally, please improve the figure resolution so that the text and traces are not blurry.

We apologise for the blurriness of the figure. We have now resolved this. The figure is for illustrative purposes and reports the chronologies for the original studies. We have not developed new chronologies for the records (as undertaken by Hoffman et al and Capron et al). Instead, we have used the d18O minima to define a common period to derive a mean temperature.

Figure 3 – This is confusing. It looks like only the modern data were run through the drift correction. I thought the correction was applied to each LIG average.

The drift correction was undertaken using a modern ocean configuration and the temperature offset applied to the average LIG estimate for each site.

Figure 4 – I recommend plotting a third panel showing the residual between the original SST and the drift-corrected SST.

We can provide this panel if the editor would like.

Table 1 – It strikes me as odd that the DJF and JJA global SST values are both negative, whereas the mean global SST value is positive. What delineated a DJF and JJA record from the other 189 records? How much overlap is there between the 92 + 99 seasonal records and the 189 annual records?

The seasonal estimates are provided in the database. Seasonal temperature estimates are challenging to provide with confidence given the seasonal biases of proxies which are likely latitudinally-dependent. As a result we consider the annual estimates to be more reliable.

Table 2 – Similar comment as above. Specific comments (regarding the Excel file):

Sheet 1 – The spatial delineations are confusing. Why do you average > 45∘ and then also > 50∘ with only 5∘ difference? Please justify.

These estimates are to provide a measure of changes in the polar latitudes. There are considerably more records polewards of 45˚ so we included both to provide a measure of the robustness of the zonal reconstructions. This is now given in the revised manuscript.

Column H - By "Jan-Dec" do you mean annual? Just say "annual" so as not to be confused with "DJF."

Done. We apologise for the confusion.

Technical corrections:

Line 42 – "The timing and impacts. . . remain. . ." instead of "remains."

Done.

Line 47 – Better references exist for "multi-millennial duration shifts in the Earth system took place in the past." The ones used here appear to mostly be about Anthropocene/ future tipping points.

Done. We have replaced with more appropriate references.

Line 51 – Can you provide a reference for 129,000-116,000 years ago, if it is elsewhere defined? Otherwise state it is the authors' definition.

Done. The reference is from Dutton et al. (2015, Science).

Line 56 – Global Mean Sea Level should not be capitalized.

Done.

Line 57 – There are better references for the observation of abrupt shifts in regional hydroclimate during the last interglacial than Thomas et al. 2015. Why not just cite cave record papers (Wang et al., 2008;Cheng et al., 2016), for example?

Done.

Line 58 – Buizert 2014 is not about $CO_2$. Kohler 2017 is partly, but why not cite the original data? (Petit et al., 1999;Barnola et al., 1987) or (Bereiter et al., 2015) for the most recent compilation of $CO_2$ ice core data.

This is correct but Buizert et al. do report $CO_2$ measurements from Taylor Dome. However, we have included these other references.

Line 61 – Provide references for "considerable debate" about the contribution of sources to sea level rise.

Done.

Line 74 – Cite also (Hoffman et al., 2017).

Done.

Line 80 – Sea-Surface Temperature should not be capitalized.

Done.

Line 83 – Can you move the Mercer 1978 reference to somewhere in the middle of the sentence? At the end of the sentence it looks like it is a reference for the Paris Climate Agreement.

Done.

Lilne 117 – Does "maximum" refer to the average of the first 5kyr? I recommend changing the wording because "maximum" can be interpreted here that your means are upper limits.

This is a fair point and we have changed.

Line 121-123 – I don't think Figure 3 should be referenced here, as it doesn't really relate to what is said in the sentence.

Done.

Line 125-129 – Again the use of the word "maximum" could be misunderstood to mean you only used the highest values in the datasets, especially on line 126.

Done.

---

## Author Comment (AC4) · 15 Sep 2020

**Response to Reviewers Comments (essd-2019-249)**

**REVIEWER #3 (JEREMY HOFFMAN)**

Turney et al. have compiled the most comprehensive data base of sea-surface temperatures spanning the last interglaciation (LIG) to date. Their results support the conclusions of several recent studies in important ways, even given their (novel) attention to potentially confounding effects present within SST reconstructions from planktonic sources (their "ocean drift") that were largely unaddressed in previous LIG work.

Understandably there has been considerable attention to the LIG as it can serve to assess the sensitivity of important Earth systems (such as the cryosphere, which was considerably smaller than at present due to higher insolation and warmer global tem- peratures) to natural climate fluctuation in recent Earth history, potentially illuminating mechanisms currently unaccounted for or underestimated in present-day climate models.

Having a "living repository" of LIG datasets from the marine realm will do well to improve future (and ongoing) LIG model-data comparisons, as is highlighted by the authors. The accompanying article is appropriate to support the publication of this dataset. The dataset is highly useful, unique in its comprehensive nature, and functionally complete. This dataset is of extremely high quality.

We were very surprised to receive this review after the completion and closure of the review process but thank the reviewer for their opening comments.

However, Turney et al. add only marginally to the existing story about total LIG warming amplitude relative to recent climatology (their uncertainties on a global anomaly overlap with basically all previous work!) and, by their chosen study design, can't add anything to the discussions ongoing about rates, extents, and locations of warming or sea-level change at particular times within the LIG. These stories have recently been borne a bit more out of work in modeling (Clark et al., 2020, Nature - referenced below) and a new ice-core based SST reconstruction (Shackleton et al., 2020, Nature Geoscience).

We are sorry to read the reviewer's comments. There is considerable work still to be completed for understanding the impact of Last Interglacial warming on the Earth system. Here we report new innovations that complement previous work. This work includes several contributions including a study into the potential role of ocean drift in reconstructing Last Interglacial temperatures, the development of a robust reconstruction of mean temperatures, the largest yet published network of quantified sea surface temperatures, and an analysis of published seasonal SSTs. The papers cited by the reviewer are important but were both published after our manuscript was submitted. In the revised manuscript we now discuss both of these studies. The paper by Clark et al provides an important analysis on the possible drivers of ice sheet melt but unfortunately restricts their model simulations of ocean temperatures to Termination 2. Here the model output suggests smaller temperatures than proxy data, highlighting the importance of extending the reconstruction further back in time. To help meet the need for future proxy-model comparisons, we have expanded on the submitted manuscript by generating late Marine Isotope Stage 6 SST estimates for records polewards of 40˚. These provide the first quantified estimates of the magnitude of the

warming from the penultimate glaciation in key ocean sectors. We are now able to recognise warming patterns in different ocean sectors. The resulting figure is provided below.

[Figure]

Figure showing the sea surface temperature increase from late Marine Isotope Stage 6 through to the maximum values reported in the early Last Interglacial. Most notably, where records are available, the greatest warming can be seen in the northeast Atlantic and south Atlantic, suggesting Greenland and the West Antarctic ice sheets would have been particularly vulnerable to warming in the early interglacial. We hope these new data may help with future coupled ocean-ice sheet modelling projects. The study by Shackleton et al. (2020) is described at length in the other rejoinders but will also be discussed (see other responses for more fuller consideration of our new analyses in respect to Shackleton et al.).

I am curious how the authors can work on an update to the manuscript that incorporates more discussion of the understanding of intra-LIG variability in sea level, temperature, and other variables, and as such, work to clearly justify just why the multi- millennial, LIG-long averages that they have generated help us to better understand those variables or model outputs. Are there modeling studies planned (lig127k PMIP?) that they can point to that would be targets for comparison with their new reconstruction? If the main SST magnitude conclusions aren't different from previous work, and the work can't resolve anything particularly new within the LIG time period, maybe the effort of the paper should simply focus on updating the maximum possible thermosteric component of LIG sea level and make that the centerpiece of the analysis?

The reviewer has correctly identified this is indeed the main objective of the study(!): to determine the contribution of ocean warming to thermosteric sea level rise. This was (and remains) the title of the manuscript: A global mean sea-surface temperature dataset for the Last Interglacial (129-116 kyr) and contribution of thermal expansion to sea-level change. We have now made explicit statements through the manuscript that we are not aiming to resolve millennial and centennial-scale variability given the considerable challenges of meaningfully resolving the timescale of many published records (as this reviewer has demonstrated).

Specific comments

Lines 188-197 – Are the ocean drift correction calculations estimated using the HadISST data used to calculate the anomalies from climatology as well? How are these "life trajectory" SST averages (which presumably have some sort of standard deviation or variance across space/time) then incorporated into the SST reconstruction uncertainty? Addressing this additional source of uncertainty in the SST estimates may further complicate the story that arises from the drift-corrected SSTs, but perhaps maybe only subtly. This might be worthwhile discussing or exploring in a couple of particular locations, especially those where the signals due to drift correction are large. I would suspect that as these areas have large SST gradients themselves that estimating an "average" SST across their lifetime/drift might generate some additional uncertainty in the estimated anomaly.

The temperature drift for the contemporary ocean is derived from the eddy-resolving ocean model, the Japanese Ocean model For the Earth Simulator or OFES. This temperature offset was then taken off the reconstructed SST values for each site. As the reviewer correctly identifies, there is more work to be undertaken investigating the impact of drift on the calibration of individual organisms into temperature, the role of differential lifespans and

settling rates etc. but that is beyond the scope of this study. We hope our work will provide a future focus for reconstructing ocean temperatures incorporating the effects of drift.

Lines 63-68 – please add Clark, P.U., He, F., Golledge, N.R. et al. Oceanic forcing of penultimate deglacial and last interglacial sea-level rise. Nature 577, 660–664 (2020). https://doi.org/10.1038/s41586-020-1931-7 to references about ice sheet modeling during this time period, as well as amounts from particular reservoirs/sources of sea-level rise. Given these recent estimates of intra-LIG sea-level change (citations within), what does this "maximum" LIG thermosteric component tell us?

We have now expanded our discussion to include Clark et al. This was published after our study was submitted to the journal and is an important contribution to the field, exploring the impact of transient changes. We have made explicit that the maximum early LIG temperature provides an upper limit on the contribution of thermosteric sea level and that later in the interglacial, the contribution was negligible. This database implies a more substantial contribution from polar ice sheets than previously supposed, particularly later in the interglacial, something we hope will be of value to the community who wish to explore ice sheet contributions to high sea-level in the interglacial.

Discussion of the LIG-long averages and addressing the small specific considerations would, in my mind, improve the clarity of this largely incremental - however important! - addition to the body of LIG SST knowledge. I thank the authors for the opportunity to comment and look forward to reading an updated draft of the manuscript.

---

## Author Response (AR2)

**Professor Chris Turney**
**ARC Laureate Fellow**

October 2020

Dear Giuseppe

It is with great pleasure that we resubmit our data descriptor paper for *Earth System Science Data* entitled '**A global mean sea-surface temperature dataset for the Last Interglacial (129-116 kyr) and contribution of thermal expansion to sea-level change.**'

Thanks so much for all your help and advice during the revision process. Please find attached a revised version of the manuscript incorporating the two changes you requested: we have clarified what 'ka' means in the Abstract (we have now restated 'the first five millennia') and we have removed the parenthesis (sorry for this oversight). We have also added the reference for the source of the map in Figure 7 as requested by Svenja and spotted a few other gremlins in the text which we have tidied up. The attached document has all the marked changes.

We hope you like the revised version and thanks once again.

Yours sincerely,

Professor Chris Turney

**14C Carbon-Cycle Facility**
University of New South Wales
Sydney, NSW, Australia, 2052
Tel: +61 2 9385 8647 Fax: +61 2 9385 8969
Email: c.turney@unsw.edu.au

[revised manuscript text omitted]

Inserted Cells

| Page 16: [7] Inserted Cells | Chris Turney | 21/10/2020 19:45:00 |
|---|---|---|

Inserted Cells

| Page 16: [8] Deleted | Chris Turney | 21/10/2020 19:45:00 |
|---|---|---|

| Page 16: [9] Deleted Cells | Chris Turney | 21/10/2020 19:45:00 |
|---|---|---|

Deleted Cells

| Page 16: [10] Formatted | Chris Turney | 21/10/2020 19:45:00 |
|---|---|---|

Font: Not Bold

| Page 16: [11] Formatted | Chris Turney | 21/10/2020 19:45:00 |
|---|---|---|

Font: Not Bold

| Page 16: [12] Formatted | Chris Turney | 21/10/2020 19:45:00 |
|---|---|---|

Font: Not Bold

| Page 16: [13] Formatted | Chris Turney | 21/10/2020 19:45:00 |
|---|---|---|

Font: Not Bold

| Page 16: [14] Formatted | Chris Turney | 21/10/2020 19:45:00 |
|---|---|---|

Font: Not Bold

| Page 16: [15] Inserted Cells | Chris Turney | 21/10/2020 19:45:00 |
|---|---|---|

Inserted Cells

| Page 16: [16] Deleted Cells | Chris Turney | 21/10/2020 19:45:00 |
|---|---|---|

Deleted Cells

| Page 16: [17] Inserted Cells | Chris Turney | 21/10/2020 19:45:00 |
|---|---|---|

Inserted Cells

| Page 16: [18] Inserted Cells | Chris Turney | 21/10/2020 19:45:00 |
|---|---|---|

Inserted Cells

| Page 16: [19] Inserted Cells | Chris Turney | 21/10/2020 19:45:00 |
|---|---|---|

Inserted Cells

| Page 16: [20] Deleted Cells | Chris Turney | 21/10/2020 19:45:00 |

Deleted Cells

| Page 16: [21] Inserted Cells | Chris Turney | 21/10/2020 19:45:00 |

Inserted Cells

| Page 16: [22] Inserted Cells | Chris Turney | 21/10/2020 19:45:00 |

Inserted Cells

| Page 16: [23] Inserted Cells | Chris Turney | 21/10/2020 19:45:00 |

Inserted Cells

| Page 16: [24] Inserted Cells | Chris Turney | 21/10/2020 19:45:00 |

Inserted Cells

| Page 16: [25] Inserted Cells | Chris Turney | 21/10/2020 19:45:00 |

Inserted Cells

| Page 16: [26] Deleted Cells | Chris Turney | 21/10/2020 19:45:00 |

Deleted Cells

| Page 16: [27] Deleted Cells | Chris Turney | 21/10/2020 19:45:00 |

Deleted Cells

| Page 16: [28] Inserted Cells | Chris Turney | 21/10/2020 19:45:00 |

Inserted Cells

| Page 16: [29] Inserted Cells | Chris Turney | 21/10/2020 19:45:00 |

Inserted Cells

| Page 16: [30] Inserted Cells | Chris Turney | 21/10/2020 19:45:00 |

Inserted Cells

| Page 16: [31] Inserted Cells | Chris Turney | 21/10/2020 19:45:00 |

Inserted Cells

| Page 16: [32] Deleted Cells | Chris Turney | 21/10/2020 19:45:00 |

Deleted Cells

| Page 16: [33] Deleted Cells | Chris Turney | 21/10/2020 19:45:00 |

Deleted Cells

| Page 17: [34] Deleted | Chris Turney | 21/10/2020 19:45:00 |